**Brief Communication**

# quantms: a cloud-based pipeline for quantitative proteomics enables the reanalysis of public proteomics data

Chengxin Dai[1,2,10], Julianus Pfeuffer[3,10], Hong Wang[1], Ping Zheng[1], Lukas Käll [4], Timo Sachsenberg [5,6], Vadim Demichev [7], Mingze Bai[1,2], Oliver Kohlbacher [5,6,8] & Yasset Perez-Riverol [9] ✉

The volume of public proteomics data is rapidly increasing, causing a computational challenge for large-scale reanalysis. Here, we introduce quantms (https://quant,ms.org/), an open-source cloud-based pipeline for massively parallel proteomics data analysis. We used quantms to reanalyze 83 public ProteomeXchange datasets, comprising 29,354 instrument files from 13,132 human samples, to quantify 16,599 proteins based on 1.03 million unique peptides. quantms is based on standard file formats improving the reproducibility, submission and dissemination of the data to ProteomeXchange.

In recent years, the field of proteomics has seen unprecedented growth in publicly available datasets, with a trend toward studies that analyze a more substantial number of samples. As of December 2023, the number of public datasets stored in the PRIDE database[1] exceeded 25,000, including a remarkable increase in large datasets containing more than 100 instrument files, from 100 in 2014 to 4,435 submissions in 2024. In parallel, a range of transformative improvements in proteomic data processing software has enabled a deeper and more precise look into the proteome. Reprocessing old data with such new tools, therefore, yields additional biological and biomedical insights[2,3]. However, the increased size of individual datasets presents a significant computational bottleneck, making it challenging to reanalyze large experiments on conventional workstations. The automated analysis of publicly accessible quantitative proteomics data is further limited by the lack of metadata that characterizes the phenotypes, the samples and the instrument operation. Although some of these challenges are tackled in earlier studies[4–6], many research groups still cannot perform automated large-scale quantitative analysis in the cloud and on distributed architectures. To address this challenge, the field requires scalable bioinformatics solutions that leverage sample metadata to automatically quantify peptides and proteins, perform absolute or differential-expression analysis and provide extensive quality control output.

Here we introduce quantms (https://quantms.org), an open-source cloud-based pipeline for massively parallel proteomic data reanalysis. It supports three major types of experiment—data-dependent acquisition label-free (DDA-LFQ), isobaric tandem mass tag (TMT)-based (DDA-plex) and data-independent acquisition (DIA-LFQ)—and is highly flexible and modular to accommodate the diversity of quantitative proteomics approaches. To enable traceable and reproducible analysis, quantms is entirely based on standardized open file formats and reproducible execution environments, adhering strictly to FAIR (findability, accessibility, interoperability and reusability) principles[7]. The pipeline is fully documented following nf-core guidelines[8].

A quantms analysis starts with the instrument files in the standard mass spectrometer format (mzML) and the protein sequence database

[1]Chongqing Key Laboratory of Big Data for Bio Intelligence, Chongqing University of Posts and Telecommunications, Chongqing, China. [2]State Key Laboratory of Proteomics, Beijing Proteome Research Center, National Center for Protein Sciences (Beijing), Beijing Institute of Life Omics, Beijing, China. [3]Algorithmic Bioinformatics, Freie Universität Berlin, Berlin, Germany. [4]Science for Life Laboratory, School of Engineering Sciences in Chemistry, Biotechnology and Health, KTH Royal Institute of Technology, Stockholm, Sweden. [5]Department of Computer Science, Applied Bioinformatics, University of Tübingen, Tübingen, Germany. [6]Institute for Bioinformatics and Medical Informatics, University of Tübingen, Tübingen, Germany. [7]Charité—Universitätsmedizin Berlin, Berlin, Germany. [8]Institute for Translational Bioinformatics, University Hospital Tübingen, Tübingen, Germany. [9]European Molecular Biology Laboratory, European Bioinformatics Institute, Wellcome Genome Campus, Cambridge, UK. [10]These authors contributed equally: Chengxin Dai, Julianus Pfeuffer. ✉e-mail: yperez@ebi.ac.uk

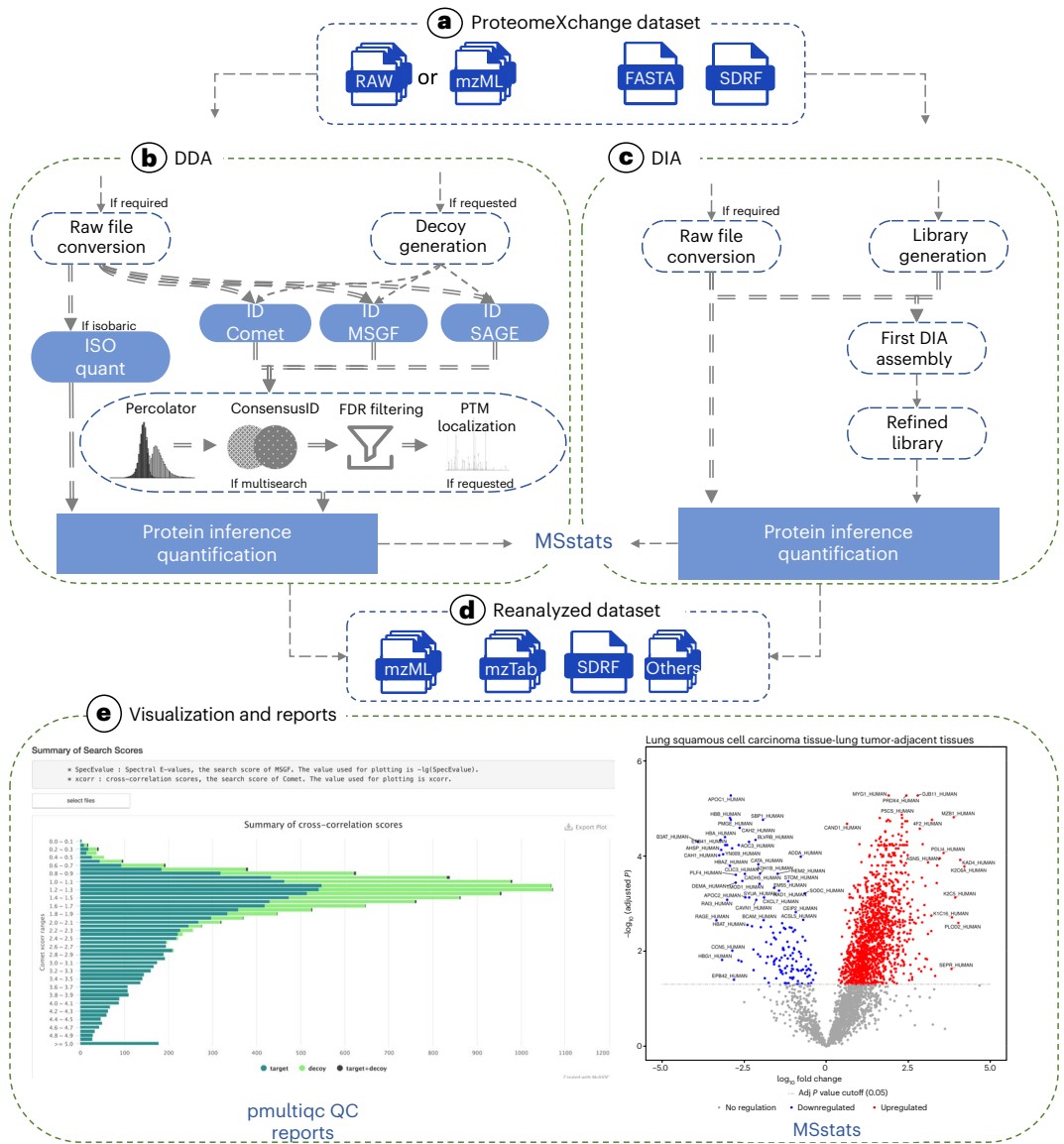

**Fig. 1 | quantms data analysis workflow and components. a**, ProteomeXchange dataset input formats for quantms for reanalysis, including the SDRF, the instrument files and the FASTA protein sequence. **b**, DDA workflow includes three major steps: (1) peptide identification (using multiple search engines and percolator for boosting the number of identifications), (2) FDR control at PSM and peptide level and (3) protein quantification for both TMT and label-free approaches. **c**, DIA workflow including the parallelization of the peptide identification and quantification step using the DIA-NN tool. **d**, Standard file formats and other supported output files from the data analysis enable automatic submissions to ProteomeXchange. **e**, Additional components of quantms workflow include pmultiqc for quality control reports and MSstats integration for differential-expression analysis of the main three approaches TMT, LFQ-DDA and DIA. The significance is calculated by a linear mixed-effects model in MSstats, and P values are adjusted by the Benjamini–Hochberg method. QC, quality control.

(Fig. 1a). The workflow uses the sample and data relationship format (SDRF)[9], to ensure the execution of workflow modules with all relevant internal parameters, including the sample variables under study and mass spectrometry-related parameters. The quantms pipeline branches into three subworkflows for DDA-LFQ, DDA-plex (Fig. 1b), and DIA experiments (Fig. 1c). Unlike conventional desktop tools such as MaxQuant, pFind[10], MSFragger[11] or ProteomeDiscover, quantms automatically distributes computation using the nextflow workflow engine[12] on one or more computers, depending on the number of instrument files and samples. To parallelize the steps that can be performed independently, the workflow streams each instrument file as annotated in the SDRF tab-delimited file to individual nodes of the computing infrastructure, such as a cloud or high-performance computing (HPC) cluster. In the final step, quantms aggregates

processed data to infer proteins, estimate protein false-discovery rates (FDRs) using the pickedFDR[13], and quantify proteins. All subworkflows export the final results into the mzTab standard format facilitating the submission of the results to ProteomeXchange (Fig. 1d). All these analyses run automatically, fully reproducibly, and without manual intervention. quantms is integrated with MSstats[14] and a Python tool (pmultiqc, https://github.com/bigbio/pmultiqc) enabling the differential-expression analysis and the generation of quality control reports (Fig. 1e). quantms uses a variety of open-source tools, including but not limited to OpenMS[15], MSstats[14], LuciPHOr2 (ref. 16), SAGE[17] and DIA-NN[18], across various stages of its workflow.

We extensively benchmarked quantms in comparison to MaxQuant on DDA-LFQ (Supplementary Notes 1 and 2)[19] and DDA-plex datasets (Supplementary Note 3). MaxQuant has been previously used for

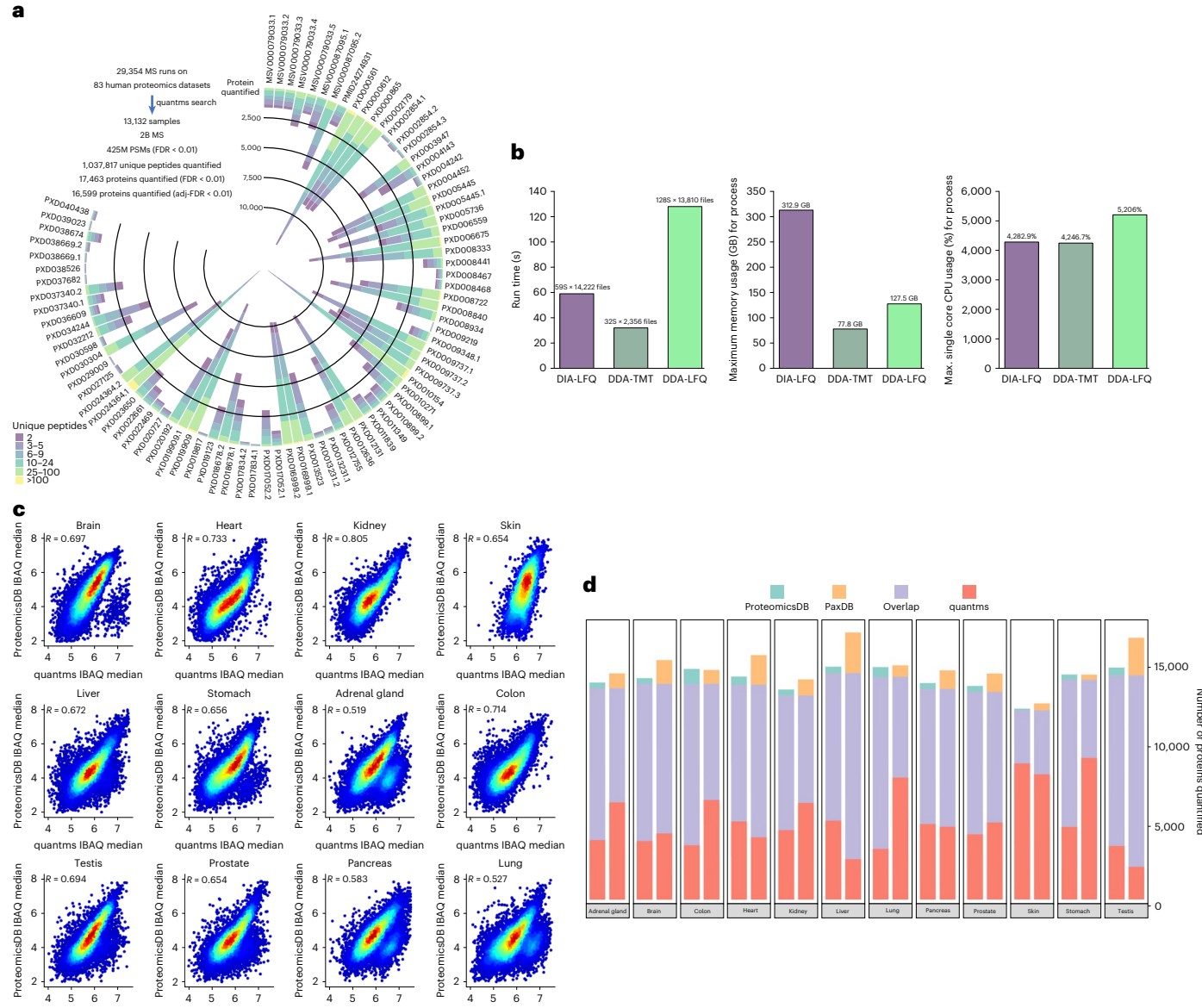

**Fig. 2 | Results of the reanalysis of 83 human public datasets from ProteomeXchange. a**, Distribution of quantified proteins by ProteomeXchange dataset, including the number of proteins with two, three to five and more than six unique peptides. **b**, Distribution of resources (CPU, memory and runtime) for the three major approaches (TMT, DDA-LFQ and DIA-LFQ), and the number of files analyzed for each approach. **c**, Correlation for 12 human tissues of the IBAQ values for the proteins quantified with quantms versus ProteomicsDB IBAQ. **d**, Overlap and unique proteins quantified in ProteomicsDB, PaxDB and quantms reanalysis.

public data reanalysis as a popular tool for intensity-based quantitation by ProteomicsDB[20], MassIVE.quant[5] and ExpressionAtlas reanalyses[1]. In summary, quantms can quantify a higher number of proteins compared with MaxQuant for all datasets with the same accuracy (lower coefficients of variation); however, for low concentrations, quantms underestimated the true fold changes (Supplementary Note 2 and Supplementary Fig. 3). In terms of scalability and performance, major differences are observed between MaxQuant and quantms. When the number of instrument files and samples grows (over 1,000 ms runs) quantms can perform 40 times faster than MaxQuant (Supplementary Note 4). quantms benefits for the parallelization and distribution of MS runs in some of the processing steps (peptide search, percolator, multiple search engine merge), decreasing the time to process big submissions. In addition, we benchmarked the DIA workflow using the dataset PXD026600 (Supplementary Note 5) and found quantms can accurately quantify spike-in ubiquitin–proteasome system (UPS) proteins on different concentrations.

In addition to benchmarking multiple datasets from the three main workflows (DDA-plex, DDA-LFQ, DIA-LFQ), we reanalyzed five public single-cell datasets from ProteomeXchange (Supplementary Note 6). While the single-cell datasets in PRIDE Archive account only for 0.38% of public proteomics data (99 datasets); the numbers continue to increase. In the PXD016921 dataset, quantms increased by 6% the number of quantified proteins compared with original results and achieved lower false-discovery identifications compared to MaxQuant. The same was observed in PXD024043 (Supplementary Note 6 and Supplementary Tables 6 and 7). The principal component analysis applied in PXD023904 shows distinct groups for classes 4 and 6, which is consistent with the original research (Supplementary Note 6 and Supplementary Fig. 21b).

To demonstrate quantms performance and scalability, we analyzed 118 human datasets, 35 differential-expression and 83 intensity-based absolute quantification (IBAQ) datasets (Supplementary Note 7 and Supplementary Table 8) based on specific dataset selection, peptide

identification, quantification and quality control rules (Supplementary Note 8 and Supplementary Table 9). quantms obtained more differentially expressed proteins compared to the original analysis in PXD030881, and these reported by only quantms were also detected as differentially expressed genes in two independent transcriptome studies (Supplementary Note 7 and Supplementary Fig. 22). The absolute (IBAQ) quantification datasets include 83 public human datasets, 13,132 samples and 29,354 instrument files (https://quantms.org/datasets). Among these datasets, we reanalyzed multiple large-scale human studies including PXD000561, PXD000865, PXD010154, PXD016999 and PXD030304. For all the DDA studies, two search engines were used (Comet and MSGF+), stringent FDR at 1% was applied at peptide spectrum match (PSM) and protein level at the dataset level, and at least two unique peptides were needed to quantify a protein. In 44 projects out of 65 (projects in which we were able to find and compare the original number of quantified proteins), quantms successfully quantified more proteins than the original analysis, and the median number of proteins quantified by only quantms was 611 (Supplementary Note 7 and Supplementary Table 8). Figure 2a shows the number of unique peptides for the 16,599 quantified. Figure 2b shows that the LFQ-DDA approach consumes more memory and central processing unit (CPU) resources than the DIA-LFQ and TMT workflows, with the DIA workflow demonstrating lower memory usage and CPU processing time despite analyzing more files. From this number, 16,270 corresponds were quantified in experiments from normal tissues, and 11,374 in cell lines experiments. It is worth highlighting that 4,993 proteins were quantified in human plasma experiments, an increase of approximately 14% compared to the PeptideAtlas Plasma identification build (Supplementary Note 7 and Supplementary Table 10). The IBAQ values computed with the quantms are highly correlated for all tissues with the proteomicDB[10] (Fig. 2c). Moreover, the present study yielded more than 479 proteins not previously quantified in ProteomicsDB or PaxDB[21] (Fig. 2d), such as Q86SP6. We have successfully quantified Q86SP6 in multiple datasets across 16 tissues including the brain, heart, kidney, lung, stomach, testis, adrenal gland, liver, colon, pancreas, ovary, esophagus, spleen, placenta, gallbladder, skin, prostate (https://quantms.org/baseline/tissues?protein=Q86SP6). Cross-checking with the UniProt database, this protein only has evidence at the transcript level and the corresponding gene, as annotated in UniProt (GPR149, Q86SP6), is expressed in 19 tissues including testis and brain, as reported by the Bgee database (https://www.bgee.org/gene/ENSG00000174948). All samples were analyzed on a high-performance computer cluster (EMBL-EBI Cluster), requiring an average of 9 hours per dataset and about 1.5 minutes per instrument run on average (Supplementary Note 7 and Supplementary Table 8).

We released a combined build of all proteins quantified using IBAQ values, in addition to the independent results reanalyzed with quantms for absolute IBAQ-based and differential-expression results. We first selected the 83 datasets from IBAQ-based reanalyses and performed a heuristic global protein-adjusted FDR procedure to assess the accumulation of false positives when merging different datasets. While all the protein inference procedures we used produce protein level $q$ values, not all of them report explicit decoy proteins with associated scores. We introduced stand-in decoy protein entries for each protein list to counter this, reflecting each individual procedure's ability to report $q$ values accurately. The procedure injects stand-in decoys so that the ratio of decoy to target proteins corresponds to the target protein's $q$ values. After this operation, we merge the lists of proteins, now with stand-in decoy proteins, and sort the resulting list according to each protein's nominal $q$ values. In cases where there is more than one observation of a protein, only the protein with the lowest $q$ value was kept. We subsequently re-estimated each target protein $q$ value as the ratio of decoy to target proteins scoring as well or better than the protein. Note that this method of injecting stand-in decoys does not improve or worsen the accuracy of the $q$ values reported by each individual procedure; it merely reflects their existing ability to report $q$ values.

Finally, we applied a strict protein-adjusted FDR threshold of less than 0.01 to filter the integration results. From the original quantified proteins with IBAQ (16,336 tissues, 11,403 cell lines and 5,048 in plasma), the number of proteins that pass the 1% adjusted FDR is 16,270 in tissues, 11,374 cell lines and 4,993 in plasma. A resource has been developed to quickly retrieve the IBAQ-based expression profile of the proteins quantified with quantms (Supplementary Note 8, https://quantms.org/baseline).

The quantms.org data constitute a resource to retrieve protein expression profiles from multiple tissues, diseases and cell lines. For every peptide and protein expression value, sample ontology-based annotations are provided. We anticipate these annotated data will prompt the creation of other resources that integrate protein expression profiles with other omics types and the development of new algorithms (for example artificial intelligence-based tools) that use the data to predict protein coexpression networks or tissue specificity expression. The quantms repository will continue releasing reanalyses of public proteomics datasets including protein variation and post-translational modification reanalysis.

quantms not only allows data processing of three different major quantification approaches, but also automates the deployment and installation of the tools used by the workflow, and converts all the output formats to standard file formats improving the reproducibility, portability and deposition of the data to PRIDE and ProteomeXchange. It also supports direct quantification reprocessing of any publicly available dataset in ProteomeXchange, in any cloud or HPC computer infrastructure. Finally, quantms is a modular and open-source workflow that enables the inclusion and extension of new (sub)workflows and pipelines for proteomics data processing. Additional documentation about the workflow, the parameters and examples can be found at https://quantms.readthedocs.io/en/latest/.

## Online content

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

## Methods

### DDA peptide identification

All branches of the workflow start with parsing the SDRF and additional user-specified options to split input files by their acquisition and labeling type, and to check and infer necessary parameters. For both LFQ and plex workflows, input files are then potentially converted into mzML and indexed. The peptide identification step for DDA-LFQ and DDA-plex approaches is shared in quantms, and three search engines are supported: SAGE, Comet and MSGF+. While multiple search engines such as MSFragger[11] and pFind[10] can manage to process large datasets in personal computers; they cannot be easily integrated into cloud open-source infrastructures. However, the novel and fast search engine SAGE[17] can be used for large-scale experiments where cluster resources are limited. These tools can be used separately or in tandem to increase the number of identifications by 5% on average (Supplementary Note 1). The workflow offers a distribution-fitting approach (reminiscent of PeptideProphet) and Percolator as methods to calculate a posterior (error) probability for each PSM. Then, the ConsensusID tool combines the PSMs from multiple search engines into a final score for each PSM. After ConsensusID, file-wide PSM-level $q$ values are taken from Percolator or calculated according to OpenMS' target-decoy strategy based on the output probabilities. The workflow performs protein inference using multiple algorithms two algorithms (Bayesian approach or aggregation) and FDR filtering using pickedFDR[13], with the same underlying algorithms as in the LFQ branch. For posttranslational modification studies, the LuciPHOr2 tool[16] can be used to compute a site-level localization score and the associated false localization rate.

### DDA-LFQ protein quantification

Two methods are available for label-free peptide–protein quantification: spectral counting and intensity-based quantification. We developed a tool proteomicsLFQ as part of the OpenMS framework[15] to perform LFQ-based quantification. For intensity-based quantification, proteomicsLFQ uses a hybrid quantification strategy that combines targeted extraction of elution profiles based on the precursors of identified peptides with an untargeted, averaging model-based feature detection approach. Chromatographic retention time alignment leverages the sample fraction annotation from the experimental design file to reduce chromatographic shifts between corresponding fractions in different instrument files. If match between runs is applied, peptide annotations are transferred from identified peptides in one run to unidentified features. An optional quantification step aims to fill the remaining missing quantitative values by running a targeted extraction based on peptide precursors that have been quantified successfully in most runs. Quantified peptides and inferred proteins are written to standardized mzTab format, MSstats and Triqler output for statistical downstream analysis.

### DDA-plex protein quantification

quantms quantification of isobaric-labeled peptides and proteins starts by reading the DDA peptide identification results into the OpenMS tool IsobaricAnalyzer. Using isotope correction matrices, this tool extracts and normalizes reporter ion intensities from MS2 and MS3 spectra. quantms currently supports 4- and 8-plex iTRAQ labeling, as well as TMT 6-, 10-, 11-and 16-plex. After protein inference and quantification, the results are again stored in standardized output formats and forwarded to downstream analysis. Three gold-standard datasets previously evaluated by TMT quantification tools were used to benchmark quantms (Supplementary Note 3). In all benchmarks, quantms performs comparably to MaxQuant and the other tools used for quantification, such as ProteomeDiscover or IsoProt (Supplementary Note 3, dataset PXD005486). In addition, we evaluated the dataset PXD007683, a two-proteome mixture in known concentrations analyzed using TMT and LFQ approaches. For both approaches, quantms quantified more proteins than MaxQuant, and both tools separated human and yeast proteins equally well (Supplementary Note 3).

### DIA protein identification and quantification

For DIA data analysis, quantms parallelizes the DIA-NN tool[18], distributing the multiple steps that DIA-NN performs on a dataset across compute nodes (Fig. 1c). The first step of the pipeline converts the protein sequence database (FASTA) into an in silico-predicted spectral library. Each instrument file in mzML is then searched against this library (First DIA assembly), resulting in a set of precursors identified. A full library of identified precursors is then created by merging all the individual searches (experimental library). A final fast identification/quantification step runs in one single node, where all the MS runs are searched against the merged experimental library (Fig. 1d). We evaluated the DIA workflow on the dataset PXD026600, an *Escherichia coli* sample with UPS1 proteins spiked in different concentrations (Supplementary Note 5). The workflow achieved nearly perfect performance (quantified all 48 UPS proteins) at four high concentrations. In addition, in most concentrations, the workflow achieved a perfect distinction between the two classes compared, namely UPS1 proteins (differentially expressed) and *E. coli* proteins (fixed background), but the accuracy naturally drops for lower concentrations, due to fewer identifications achieved and noisier quantification (Supplementary Note 5).

### Downstream analysis and quality control

MSstats and quantms are fully integrated for differential-expression data analysis. The workflow generates input for the MSstats R package, and if differential-expression analysis is performed, the MSstats plots and output files are automatically produced. MSstats was selected after benchmarking MSstats and other R packages with quantms for multiple LFQ datasets[19]. Factor values and/or conditions, and biological and technical replicates under study are translated from the original SDRF (provided as input format) to MSstats columns. In cases where multiple SDRFs are being used to study multiple conditions or factor values, the pipeline will reuse steps that have already been executed with no changes in parameters, and only execute the step that differs due to the SDRF being used (such as the quantification step in proteomicsLFQ). Users can automatically perform the differential-expression analysis using MSstats (https://quantms.readthedocs.io/en/latest/msstats.html). The workflow will detect whether the pipeline is LFQ (DIA or DDA) or TMT and will use the corresponding MSstats package (MSstats or MSstatsTMT) accordingly. The MSstats step will generate by default a list of plots, including a volcano plot, quality control plot and comparison plot (for example, http://ftp.pride.ebi.ac.uk/pub/databases/pride/resources/proteomes/differential-expression/PXD004683/msstatstmt/). Configurable parameters for MSstats data processing step includes the summary method, the log fold-change threshold and so on.

To ensure high-quality data, we developed pmultiqc (https://github.com/bigbio/pmultiqc), which is part of the quantms tool ecosystem (Fig. 1e and Supplementary Note 9). pmultiqc generates a quality control report for each analyzed dataset, using the mzTab, SDRF and other intermediate files. The report includes different plots that display the number of peptides identified per protein, the distribution of PSM posterior error probabilities and search engine scores or the MS2/MS3 identification rate.

### Portability and deployment

All quantms tools are available as versioned BioConda packages and BioContainers and the workflow has been developed using the nextflow and nf-core[8] guidelines enabling compatibility with an ecosystem of infrastructures including Amazon Web Services, Google Cloud Platform, Kubernetes and HPC clusters (Supplementary Note 10). Owing to its implementation as an nf-core/nextflow workflow, quantms allows resuming failing process executions as well as reallocation of resources

(for example, memory and CPU), depending on the demands of the tool and workflow monitoring.

### Interoperability and ProteomeXchange support

quantms processing steps are based on standard file formats. The input formats are SDRF and mzML and the main result files are exported into mzTab. To export DIA and DDA results into mzTab, new controlled vocabulary terms and external reference files were introduced. In addition, the pipeline automatically generates other file formats that can be used for downstream analysis, such as MSstats and Triqler inputs. Results from quantms can be readily submitted to PRIDE and ProteomeXchange as COMPLETE submissions.

### Reporting summary

Further information on research design is available in the Nature Portfolio Reporting Summary linked to this article.

## Data availability

The datasets reanalyzed in the present study can be searched on the quantms web page (https://quantms.org/datasets). In addition, all the results can be found in the PRIDE database FTP (http://ftp.pride.ebi.ac.uk/pub/databases/pride/resources/proteomes/). Source data are provided with this paper.

## Code availability

All software, algorithms and tools are available on GitHub: quantms at https://github.com/bigbio/quantms and pmultiqc at https://github.com/bigbio/pmultiqc. The full documentation of quantms is available at https://quantms.readthedocs.io/en/latest/.

## Acknowledgements

Y.P.-R. was funded by the EU H2020 project EPIC-XS (grant no. 823839), Wellcome grants (nos 208391/Z/17/Z, 223745/Z/21/Z) and EMBL core funding. M.B. and C.D. were funded by the National Key Research and Development Program of China (grant no. 2018YFA0507504). V.D. was supported by the Federal Ministry of Education and Research (BMBF), as part of the National Research Initiatives for Mass Spectrometry in Systems Medicine ('MSCoreSy'), under grant agreement no. 161L0221.

## Author contributions

C.D., J.P. and Y.P.-R. developed the quantms workflow. H.W., C.D., J.P. and Y.P.-R. developed the pmultiqc library and web application, P.Z., H.W., C.D. and Y.P.-R. developed the quantms.org web page to present the results of the quantitative analysis. T.S. and J.P. developed the algorithms and tools in OpenMS for DDA-plex and LFQ-DDA workflow. V.D. contributed to the development of the algorithm parallelization of the DIA-NN tool and the LFQ-DIA workflow. C.D., J.P. and Y.P.-R. performed the annotations of the datasets and the data analysis. L.K. designed and developed the stand-in decoy. C.D., J.P., T.S., V.D., M.B., O.K. and Y.P.-R. wrote the paper and contributed to the design of the workflow and quantms.org project.

## Funding

## Competing interests

The authors declare no competing interests.

## Additional information

**Correspondence and requests for materials** should be addressed to Yasset Perez-Riverol.

| | |
|---|---|

# Reporting Summary

## Statistics

For all statistical analyses, confirm that the following items are present in the figure legend, table legend, main text, or Methods section.

| n/a | Confirmed | |
|---|---|---|
| ☐ | ☒ | The exact sample size (*n*) for each experimental group/condition, given as a discrete number and unit of measurement |
| ☐ | ☒ | A statement on whether measurements were taken from distinct samples or whether the same sample was measured repeatedly |
| ☐ | ☒ | The statistical test(s) used AND whether they are one- or two-sided<br>*Only common tests should be described solely by name; describe more complex techniques in the Methods section.* |
| ☐ | ☒ | A description of all covariates tested |
| ☐ | ☒ | A description of any assumptions or corrections, such as tests of normality and adjustment for multiple comparisons |
| ☐ | ☒ | A full description of the statistical parameters including central tendency (e.g. means) or other basic estimates (e.g. regression coefficient) AND variation (e.g. standard deviation) or associated estimates of uncertainty (e.g. confidence intervals) |
| ☒ | ☐ | For null hypothesis testing, the test statistic (e.g. *F*, *t*, *r*) with confidence intervals, effect sizes, degrees of freedom and *P* value noted<br>*Give P values as exact values whenever suitable.* |
| ☒ | ☐ | For Bayesian analysis, information on the choice of priors and Markov chain Monte Carlo settings |
| ☒ | ☐ | For hierarchical and complex designs, identification of the appropriate level for tests and full reporting of outcomes |
| ☒ | ☐ | Estimates of effect sizes (e.g. Cohen's *d*, Pearson's *r*), indicating how they were calculated |

*Our web collection on statistics for biologists contains articles on many of the points above.*

## Software and code

Policy information about availability of computer code

| Data collection | All data accessions are mentioned in the supplementary information of the manuscript. |
|---|---|
| Data analysis | All software, algorithms and tools are available on GitHub: quantms (https://github.com/bigbio/quantms), pmultiqc (https://github.com/bigbio/pmultiqc). The full documentation of quantms is available at (https://quantms.readthedocs.io/en/latest/). |

For manuscripts utilizing custom algorithms or software that are central to the research but not yet described in published literature, software must be made available to editors and reviewers. We strongly encourage code deposition in a community repository (e.g. GitHub). See the Nature Portfolio guidelines for submitting code & software for further information.

## Data

Policy information about availability of data

All manuscripts must include a data availability statement. This statement should provide the following information, where applicable:
- Accession codes, unique identifiers, or web links for publicly available datasets
- A description of any restrictions on data availability
- For clinical datasets or third party data, please ensure that the statement adheres to our policy

The datasets reanalyzed in the present study can be searched on the quantms web page (https://quantms.org/datasets). In addition, all the results can be found in the PRIDE database FTP (http://ftp.pride.ebi.ac.uk/pub/databases/pride/resources/proteomes/).

## Human research participants

Policy information about studies involving human research participants and Sex and Gender in Research.

| | |
|---|---|
| Reporting on sex and gender | N/A |
| Population characteristics | N/A |
| Recruitment | N/A |
| Ethics oversight | N/A |

Note that full information on the approval of the study protocol must also be provided in the manuscript.

# Field-specific reporting

Please select the one below that is the best fit for your research. If you are not sure, read the appropriate sections before making your selection.

☒ Life sciences  ☐ Behavioural & social sciences  ☐ Ecological, evolutionary & environmental sciences

For a reference copy of the document with all sections, see nature.com/documents/nr-reporting-summary-flat.pdf

# Life sciences study design

All studies must disclose on these points even when the disclosure is negative.

| | |
|---|---|
| Sample size | We use datasets from the public domain, the sample size was define but the original submitters of the data. |
| Data exclusions | No data exclusions was implemented |
| Replication | We use datasets with multiple replicates depending on the data availability. |
| Randomization | All data was used from multiple datasets in the public databases for proteomics. |
| Blinding | We didnit select any datasets for reanalysis based on authors, laboratories, journals were the datasets |

# Reporting for specific materials, systems and methods

We require information from authors about some types of materials, experimental systems and methods used in many studies. Here, indicate whether each material, system or method listed is relevant to your study. If you are not sure if a list item applies to your research, read the appropriate section before selecting a response.

### Materials & experimental systems

| n/a | Involved in the study |
|---|---|
| ☒ | ☐ Antibodies |
| ☒ | ☐ Eukaryotic cell lines |
| ☒ | ☐ Palaeontology and archaeology |
| ☒ | ☐ Animals and other organisms |
| ☒ | ☐ Clinical data |
| ☒ | ☐ Dual use research of concern |

### Methods

| n/a | Involved in the study |
|---|---|
| ☒ | ☐ ChIP-seq |
| ☒ | ☐ Flow cytometry |
| ☒ | ☐ MRI-based neuroimaging |

