## [Peer Review File · Nature Methods]

Peer Review Information

Manuscript Title: quantms: A cloud-based pipeline for quantitative proteomics enables the reanalysis of public proteomics data.

Corresponding author name(s): Yasset Perez-Riverol

Editorial Notes: None

Reviewer Comments & Decisions:

Decision Letter, initial version:

Dear Dr. Perez-Riverol,

Your Brief Communication entitled "quantms: A cloud-based pipeline for proteomics reanalysis enables the quantification of 17521 proteins in 9,502 human samples." has now been seen by 3 reviewers, whose comments are attached. While they find your work of potential interest, they have raised serious concerns which in our view are sufficiently important that they preclude publication of the work in Nature Methods, at least in its present form.

As you will see, the reviewers raise concerns about the advance of quantms over existing tools, as well as concerns regarding its practical utility.

Should further experimental data allow you to fully address these major criticisms we would be willing to look at a revised manuscript (unless, of course, something similar has by then been accepted at Nature Methods or appeared elsewhere). This includes submission or publication of a portion of this work somewhere else. We hope you understand that until we have read the revised paper in its entirety we cannot promise that it will be sent back for peer-review.

If you are interested in revising this manuscript for submission to Nature Methods in the future, please contact me to discuss your appeal before making any revisions. Otherwise, we hope that you find the reviewers' comments helpful when preparing your paper for submission elsewhere.

Sincerely yours,
Allison

Allison Doerr, Ph.D.
Chief Editor
Nature Methods

Reviewers' Comments:

Reviewer #1:

Remarks to the Author:

Dai et al. present a software tool for a large scale proteomics data analysis including identification and quantification and statistical testing. My main comments are:

1. The motivation for this tool in the intro is that reprocessing old data with new tools "...yields additional biological and biomedical insight." The authors have not demonstrated this. The comparison to original analyses is limited to the number of proteins. Something interesting should be done with the new data from the re-analysis to show some benefit if this were to be broadly impactful work.
2. I see the main utility of such a tool being single cell proteomics where we have hundreds and thousands of files. This application is not mentioned.
3. It appears just the code is available without a public interface to use it. That means separate nodes would need to be set up for each user at their institution, requiring specialized expertise. I'm not sure how widely impactful this is if there is not a public interface.

Minor comments:

There are many typos and grammar errors throughout the text. For example in supplementary note 1 "By February 2013", and the last two sentences in the abstract.

Supplementary note 1 shows the different number of peptides from each search. Statistical tools exist to combine these outputs such as iprophet. Are the data not being combined in a statistically rigorous way? There should be another column for the combined peptide count.

Link to the report mentioned in suppl note 1 doesn't work. The link to the PXD001819 benchmark doesn't work.

Supplemental notes line 128-129: "Protein sequence databases are made available in each dataset's GitHub folder." Is this statement correct? In my experience it's rare that a dataset would store data in a github folder.

Supplemental notes line 149-150: "the number of missing values in the background proteins was higher in quantms than MQ." This is not supported by the numbers in the table where missing values are lower than MQ (2.2% for quantms versus 31% for MQ; and 4.6% for quantms versus 5.3% for MQ).

Supplemental note 1, figure 3: it says this data is from 48 proteins spiked into yeast, how can there be $N > 7,000$ where yeast only expresses about 4k proteins? Also in the second panel showing the spike-in proteins, how can there be $N > 300$ when there are 48 spike-in proteins?

Reviewer #2:

Remarks to the Author:

We think quantms is a potentially interesting tool with the novel concept of nextflow being implemented for proteomics analysis. However, there is a big question mark about whether such a tool could justify its publication at Nature Methods.

Here are some major critiques:

- All the software tools quantms integrated are not new. Also, nextflow was already published in Nature Biotech. Quantms is simply an assembly of published tools.
- It is impressive that the authors searched data from almost 10k human samples (>20k raw files) using quantms. This being said, recent fast-searching tools such as MSfragger and pFind could easily search thousands of files in a single powerful PC. Also, the present large-scale searching was based on their own HPC cluster (204 cores, Supplementary Note 4), but not on a cloud resource like AWS. It is not clear if the cost range for such a search is too prohibitive for a normal user/lab using a cloud ecosystem.
- No novel biological findings were present – it would be much more interesting if the author re-searched e.g., all the CPTAC data with quantitative outputs.
- The total number of 17k proteins identified, together with the caveat of not having a particular protein-level FDR control for the extremely large datasets, questioning such an impressive number of proteins for humans. Some benchmark work should be done by e.g., searching the data for non-human proteins or olfactory receptors (Ezkurdia et al 2014). Otherwise, such a title in NM could be misleading for the proteomics field.
- Finally, reading from the materials the author provided, we feel that the whole pipeline is not user-friendly, especially not for beginners. The troubleshooting of the command lines can be painful and troublesome. No case study was present in detail. In general, the author could consider improving features (e.g., using GUIs) to facilitate the usage.

To summarize, although we like the concept of this tool, we doubt how useful it could be for normal proteomic users and therefore the significance and novelty of this contribution.

On a separate note, it is not clear to us if quantms could support the direct, automatic procession of the PRIDE datasets (based on their IDs). If the normal users of PRIDE still need to download and copy the PRIDE datasets to a server, this can be very time-consuming (compared to a local analysis). If they do not need to, this is a highlight for quantms.

Minor:

1. The speed comparison to Maxquant was unfair due to the different HPC clusters used.
2. It is not clear how quantms will incorporate and support different versions of software tools it currently assembles- This can be very important for, e.g., DIA-NN due to the development in the field of DIA data analysis.
3. Authors should cite MSfragger and pFind with some discussions on what factors the users should consider if they want to use quantms.
4. Figure 2B (run time and max memory usage) is not helpful because the final performance is linked to the particular HPC cluster they used and thus is not informative and transferable.
5. The TMT searching parameter cannot be found at <https://nf-co.re/launch?id=1687271608e665f052d31c>
6. The link to Jupyter notebook <https://github.com/ypriverol/quantms-research/blob/main/notebooks/TMT/PXD007683Benchmark.ipynb> does not work.

Reviewer #3:

Remarks to the Author:

The authors propose an open source, cloud-based pipeline for parallel reanalysis of quantitative proteomic datasets. The authors argue that the tool is needed due to an increasingly large sizes of the experimental datasets, which makes it hard to do the re-analyses on individual workstations.

While I agree that the tool is needed, I do not see a lot of novelty in the tool itself, as it is primarily an engineering effort that connects existing well-characterized modules. However, the datasets produced by a reanalysis with the tool are indeed novel and necessary. I believe that the manuscript will make an important contribution if it goes beyond stating the fact that we can re-analyze the studies on a cloud, and convinces the community that these are in fact reanalyses that we can trust. In my opinion the current version of the manuscript doesn't go quite far enough in that direction.

Currently, all the benchmark comparisons are in supplementary notes, and detailed explorations of reanalyzed datasets are limited to 3 controlled mixtures and 2 biological investigations. It is not clear how many plexes are in the TMT datasets, it appears that it is just one. The results may be quite different for multi-flex experiments. I could not locate a detailed example of a biological DIA dataset. The remaining datasets are mentioned in the passing. I suggest that the discussion of the reanalyzed datasets is moved front and center, and more evidence of accuracy of reanalyses is given for all the datasets, not just the selected few.

I would caution against focusing too much on controlled mixtures. While they are useful, they do not have biological variation representative of the actual studies. Therefore, I suggest giving more attention to data sets from human samples.

The number of identified proteins, and the computational speed are of course important, but these are not the most important criteria in my opinion. Instead, I suggest that the authors focus on demonstrating the soundness of the quantitative analysis, and its accuracy. I have specific questions in this regard:

- How did the authors select dataset-appropriate settings at various stages of the analyses (such as choice of a database and modifications; mass resolution/accuracy; filtering out features with poor id or quant; filtering features with too much noise and too many missing values)? These criteria are dataset-specific. Is this decision automated? Or do users need to specify these parameters? What is the impact of the choice of these parameters on the quantitative accuracy? Which combinations result in more trustworthy reanalyzed datasets?

- How can the tool check that the order of the acquisitions is appropriately randomized? Can the users be warned that the results cannot be trusted due to lack of or inappropriate randomization?

- How do the authors choose the between-run normalization and correction for batch effects? For example, median normalization may not be appropriate for some controlled mixtures, or for experiments with standards. If the experiment includes standards, can/should the normalization be done with respect to these standards? These decisions are dataset-specific. Are they automated? What is the impact of the choice on the accuracy?

- How are the reanalyses made available to the community, at the intermediate and at the final stages? E.g., if one wanted to change only one module in the workflow for one of these datasets, can we start from an intermediate step, or do we need to restart the analysis from scratch? If we can start in the middle, where can we find the intermediate input files?
- Since the workflow is modular, and some modules can be substituted, could the authors illustrate the impact of this modularity on the quantitative results? I.e., how similar or different are the results when you substitute one module with another module designed for the same purpose?
- For previously published data sets, how similar or different are the results as compared to the original publications, e.g. in terms of number of proteins, features per protein, missing values, (relative) protein quantification or detection of differentially abundant proteins? What can explain the differences, if any? Which analysis is more trustworthy?

Additional questions:

What is the relationship between the annotation formats used by this framework, annotation formats in MassIVE.quant, and annotation formats for quantitative experiments in PRIDE?

What is the relationship between quantms and the reanalysis capabilities in MassIVE?

Can the intermediate analyses files be stored in MassIVE.quant?

Author Rebuttal to Initial comments

Summary

After carefully considering the points raised by the reviewers and the editor, we acknowledge that we need to improve the presentation of quantms' strengths and broad impact.

Firstly, we want to emphasise that quantms represent a significant curation effort yielding experimental design annotations of 118 datasets available now at <https://quantms.org/datasets>. One common concern from reviewers was that the workflow is not novel and only combines existing tools. Here we did a bad job explaining that one of the key technical innovations implemented in quantms is the incorporation of novel or extended tools that are aware of the experimental design throughout the data analysis and statistical downstream processing. This clearly distinguishes quantms from standalone applications or pipelines that merely combine existing ones. The second class of newly developed quality control tools like pmultiqc (example report available at http://ftp.pride.ebi.ac.uk/pub/databases/pride/resources/peptomes/single-cell-expression/PXD016921/summarypipeline/multiqc_report.html) ensure that results produced by quantms maintain a high standard for data analysis. These novel tool developments have been instrumental in achieving the level of automation we have attained, and we attribute this advancement to an updated version of the manuscript.

We also expand upon how quantms as open and free HPC/cloud analysis platforms enable fully automated reanalysis for medium to large research facilities or companies and how this effectively democratises large-scale analysis of all major MS-based proteomic methods (DDA-LFQ, DDAplex, and DIA-LFQ) - a task previously limited to a few key players in the field. In large-scale processing, obtaining a robust solution that e.g., allows resuming from error states (e.g., hardware failure) is crucial. We have included additional data reanalysis as requested by reviewers.

quantms reanalysis results discovered previously unidentified proteins indicating that results are novel. To increase confidence in the discoveries, we expand the manuscript with a global adjusted FDR estimation/filtering. To our knowledge, no major public data reanalyses have been performed of human data that included the annotations of the samples, fractions, technical and biological replicates, and the experimental design of every experiment. While reanalysing multiple tissue proteome datasets, we gain a deeper understanding of protein expression profiles at the tissue level. Building on 83 studies we obtain the most comprehensive view on tissue protein expression. In Figure 2 of the manuscript, we illustrate the results, showcasing the number of quantified proteins previously unreported in tissue-specific databases like ProteomicsDB and PaxDB. (Please see the detailed explanation of new findings on independent differential expression datasets and the absolute expression results R1.1). We are thus certain that the data alone produced by quantms has showcased its potential to uncover new knowledge from existing datasets.

At the same time, we admit that we did a poor job making the data FAIR (Findability, Accessibility, Interoperability, and Reusability). In the present version of the manuscript, we have released a web page quantms.org, including information on the workflow, the datasets reanalysed, the proteins quantified with IBAQ values on different tissues, and the link to resources such as PRIDE.

By consistent processing and making the data FAIR we will enable groups to reuse original data, sample information, peptides and protein intensities (including extensive metadata) facilitating the development of new ML/AI methods (e.g., prediction of tissue specificity or protein expression prediction, etc.). We think this is a significant achievement that can lead to breakthroughs in various fields, including proteomics and biomedical research. However, the direct validation of novel biomarkers and various differential proteins, as suggested by one reviewer, is a tremendous community effort that expands beyond the scope of this manuscript.

Finally, we want to stress that the openness of quantms fosters the testing and implementation of novel or improved methods, alternative tools, and additional workflows, contributing to a collaborative and dynamic research environment. We acknowledge the additional valuable comments from the reviewers and address them in the current version. These are the main changes included in the current version of the manuscript:

- *Extra biological examples: We present additional cases showcasing the transformative improvements quantms offers over previous analyses, utilising data we already processed.*
- *Improved presentation of absolute expression data obtained using a consistent workflow. Expression data is meticulously annotated at the sample level, a scale unprecedented in previous endeavours.*
- *We are incorporating additional data from single-cell reanalysis and the CPTAC consortium, as recommended by the reviewers.*
- *FAIR data release and web page (quantms.org).*
- *Clarification and discussion of concepts: We thoroughly address parts of the manuscript where the central concepts require further clarification or were not adequately presented in the original version.*

We believe that these actions, along with the unique aspects of quantms, will address the reviewers' concerns and significantly contribute to the field.

Reviewer #1:

Remarks to the Author:

Dai et al. present a software tool for large-scale proteomics data analysis including identification and quantification and statistical testing. My main comments are:

1. The motivation for this tool in the intro is that reprocessing old data with new tools "...yields additional biological and biomedical insight." The authors have not demonstrated this. The comparison to original analyses is limited to the number of proteins. Something interesting should be done with the new data from the re-analysis to show some benefit if this were to be broadly impactful work.

R1.1 / While it is easy to show that using novel tools to analyze old data increases the number of reported proteins, we agree that the statement about additional biological and biomedical insight may need to be put in the right context. We rephrase this part to emphasize the potential benefits of reprocessing old data with new tools while acknowledging that demonstrating additional biological and biomedical insights likely

requires further investigation. Nevertheless, we tried to make a strong point, why we believe that additional biological insight can be extracted from quantms results by expanding the comparison between the reanalysis and the original analyses beyond the number of proteins. First, we added a summary, detailed in Table 6, that illustrates notable differences between the reanalysis and the original analysis.

We selected one suitable study, PXD030881, to demonstrate that the majority of differential proteins only found by quantms reanalysis are supported by differential genes from two transcriptomic studies.

Statistical downstream analysis with the MSstatsTMT (integrated into quantms) tool detected 3381 (original) and 4301 (reanalysis) differential proteins (DEPs, adj. p -value < 0.05) [PMID 35335125]. Next, we compared the 1762 DEPs found only by quantms with two independent transcriptomics results [PMID23077249, PMID34395436]. Of the 1762 novel DEPs reported in our results 1035 differential expressed proteins were also detected as differentially expressed genes in both independent transcriptome studies. We argue that this is a strong indication that quantms results provide novel biological insights.

This example is now added to the supplementary information of the manuscript to make clear how quantms reanalysis improved the original results.

In addition, to the differential expression analyses, we quantified over 400 proteins for the first time (previously not reported in a proteomic resource as expressed in specific tissues). For example, protein P01721 protein is reported as only expressed (quantified) in the spleen tissue (databases: paxDB and proteomicsDB). However, using quantms data, we have discovered that protein P01721 is consistently quantified in multiple datasets across various tissues such as the heart, colon, prostate, pancreas, lung, kidney, and stomach (<https://quantms.org/ae/tissues?protein=P01721>). The gene expression resources have expression evidences for the corresponding gene (IGLV6-57 - ENSG00000211640) in more than 93 tissues (e.g. https://www.bgee.org/gene/ENSG00000211640?expression=anat&data_type=RNA_SEQ%2CSC_RNA_SEQ). An in PeptideAtlas (https://db.systemsbiology.net/sbeams/cgi/PeptideAtlas/GetProtein?atlas_build_id=550&protein_name=P01721&action=QUERY) the protein has been mostly identified in plasma, stomach and heart similar to the quantms quantification results.

Some proteins like Q86SP6 are not reported in PaxDB and ProteomicsDB at all. However, utilizing quantms, we have successfully quantified Q86SP6 in multiple datasets across 16 tissues including the brain, heart, kidney, lung, stomach, testis, adrenal gland, liver, colon, pancreas, ovary, esophagus, spleen, placenta, gallbladder, and prostate (<https://quantms.org/ae/tissues?protein=Q86SP6>). The corresponding gene (GPR149) is expressed in 19 tissues (https://www.bgee.org/gene/ENSG00000174948?expression=anat&data_type=RNA_SEQ%2CSC_RNA_SEQ) including testis, adrenal, and et al. Cross-checking with the UniProt database, this protein only has evidence at the transcript level. (See responses R2.4 and R3.3 for the discussion of the statistical assessment of the tissue analyses)

We hope that the comprehensive table and the exemplary examples are convincing that the large-scale reanalyses performed by quantms provides additional information likely yielding novel biological insights and serve as a valuable complement to the existing resources.

The following table (added as Supplementary Information – Table 6) summarizes the differences between the reanalyses and the original results.

Table 6: Datasets reanalyzed using quantms, (DE) differential expressed studies, (AE) intensity-based absolute expression studies.

Accession	Type	# msruns	# samples	# of MS	# PSMs	# peptides	# proteins quantified	# original peptides	# original proteins quantified	Diff Proteins	cpu runtime (hours)
PXD010271	AE	118	117	4144357	655277	69667	5442	68623		5442	92
PXD028618	DE	18	18	3636018	295437	26261	3478		1414	2064	16
PXD021394	DE	24	24	1550158	1123224	63072	6577		4644	1933	13
PXD025560	DE	203	203	28625259	5413288	88199	8523		6717	1806	181
PXD018830	DE	25	25	1422778	963529	64270	6333	28746	4617	1716	12
PXD010899.1	AE	1808	26	26814390	1391181	41042	3288	15424	1626	1662	714
PXD032212	AE	18	9	1378389	540117	37557	3138		1496	1642	52
PXD002137	DE	192	32	14938434	5189483	166811	10550	111680	9112	1438	265
PXD004684	DE	15	8	801646	340501	30074	3503	12976	2155	1348	3
PXD000672	DE	36	36	2610072	726081	25636	2971		1632	1339	173
PXD003539	DE	120	60	9385023	1647895	23413	3509	18030	2174	1335	176
PXD014943	DE	113	113	15031034	4406765	57115	5609	31952	4365	1244	176
PXD002395	DE	198	33	7779229	4765530	181082	10530	126235	9307	1223	211
PXD010154	AE	1367	38	76331811	13644346	344215	14602	277698	13413	1189	6467
PXD022992	DE	12	6	1056747	1491915	135706	9474		8455	1019	17
PXD032263	DE	8	4	684223	933494	112288	8079	77928	7089	990	10
PXD019909	AE	154	12	17118239	3223352	243137	11680	173228	10701	979	350
PXD004691	DE	224	97	13851936	3709841	26493	2956	17108	2009	947	181
PXD004873	DE	76	38	6336044	1788264	31725	3404	11787	2579	825	43
PXD030881	DE	24	10	1684376	407792	116306	8236	84448	6689	1547	44
PXD012755	AE	32	32	1842536	816445	20519	2325	8017	1640	685	14

PXD030671	DE	18	33	742449	184438	64775	4665		4051	614	22
PXD023650	AE	48	1	916155	97510	29383	2797	19508	2190	607	24
PXD008333	AE	201	8	7804654	1993476	98165	8144	83984	7609	535	122
PXD008934	AE	34	34	3637030	1133506	42867	3433	31088	2933	500	54
PXD012131	AE	312	26	50179092	4575069	166338	10231	129050	9735	496	459
PXD030304	AE	6862	2013	1135197814	241592436	118322	8941		8498	443	23352
PXD010899.2	AE	282	40	5881201	172728	16497	1506		1080	426	448
PXD008722	AE	252	21	9250425	3397589	134566	6851	107417	6436	415	239
PXD034244	AE	29	9	3855168	1088315	40832	5064		4659	405	28
PXD003947	AE	108	10	4063798	1003593	84584	5087		4727	360	46
PXD019909.1	AE	43	43	3393681	2444015	127893	9445		9140	305	15
PXD005445	AE	105	7	6657738	2816862	66690	9220		8980	240	81
PXD025864	DE	18	6	888111	318986	31954	3101	26657	2877	224	1.7
PXD012636	AE	90	9	3989225	1725188	113962	6425	110736	6243	182	120
PXD022661	AE	60	5	667548	159833	12788	1667	12096	1492	175	11
PXD008840	AE	504	84	36907863	6493137	147323	9296		9186	110	1094
PXD013231.2	AE	1549	1549	48873931	9759381	7437	562		465	97	931
PXD004242	AE	1290	58	30415188	4562256	8029	529		448	81	1363
PXD020727	AE	41	41	2980280	126333	7398	657		598	59	29
PXD020109	DE	12	4	719423	80722	7736	1217	6366	1173	44	22
PXD019817	AE	98	98	2731283	197079	3469	242		199	43	60
PXD030598	AE	435	870	214304800	2847574	3827	385		366	19	1753
PXD006675	AE	448	56	39666383	10788990	210623	11178		11163	15	2082
PXD028251	DE	51	51	4444942	1366811	7134	538		548	-10	54
PXD012574	DE	24	30	1053235	235755	47577	4867	20868	4905	-38	35
PXD020248	DE	4	12	215854	100786	34042	3458		3500	-42	4

PXD040438	AE	24	24	1293983	96783	5759	383		447	-64	17
PXD017834.1	AE	9	3	808262	24665	2716	172	2332	241	-69	9
PXD017834.2	AE	18	6	1311243	71847	5004	305		376	-71	20
PXD010429	DE	4176	348	27150819	2887185	162572	8227	158160	8300	-73	553
PXD023508	DE	80	80	23183808	157303	2805	265		380	-115	250
PXD009348	AE	168	7	4589074	375058	19466	1707	14588	1826	-119	64
PXD016999	AE	336	280	33132420	3312949	185984	10322		10442	-120	762
PXD011349	AE	55	55	1093212	344925	12061	1309		1445	-136	12
PXD029009	AE	366	366	4812900	690169	1901	161		321	-160	301
PXD038674	AE	48	16	616997	219695	13871	1230		1429	-199	13
PXD009737.1	AE	36	1	2249213	570500	153489	10477	257785	10743	-266	27
PXD008441	AE	115	115	2752279	378840	20287	1658		1929	-271	46
PXD019123	AE	27	9	1281714	423137	24714	2860	24636	3180	-320	16
PXD004683	DE	48	12	1197561	242962	64235	4903	50754	5513	-610	45
PXD006559	AE	10	72	1103416	210748	71736	5160		5816	-656	20
PXD009737.2	AE	36	1	2051259	356753	98641	7839	247234	8849	-1010	19
PXD009737.3	AE	36	1	1912707	433716	91533	8439	253038	9700	-1261	18
PXD013523	AE	96	16	4253344	1657534	65554	5855	85768	7414	-1559	58

Supp. Notes - Table 6: Summary of quantified peptides and proteins (Supplementary Notes - Table 6) from reanalyzed projects. We have included the peptides/proteins quantified in the original manuscript whenever possible. In 44 projects out of 65 (projects in which we were able to find and compare the original number of quantified proteins), quantms successfully quantified more proteins than the original analysis, as indicated by the green rows in the column "Diff Proteins." Conversely, in 21 projects, quantms quantified fewer proteins than the original analysis. Notably, in the projects where quantms yielded more quantified proteins (green rows) than the original analysis, the median number of proteins quantified by

only quantms was 611, while the median for the proteins not quantified by quantms in the 21 projects (red rows) was 136. It is important to highlight that for each independent project for a protein to be considered quantified with quantms, a stringent filter was applied: 1% FDR at PSMS level, 1% FDR at protein level, 2 unique quantified peptides per protein, more than 7 AA for each peptide quantified. These thresholds and rules were more stringent than those in most of the original analyses.

2. I see the main utility of such a tool being single-cell proteomics where we have hundreds and thousands of files. This application is not mentioned.

R1.2 / We agree with the reviewer that quantms is useful for large-scale datasets, including single-cell datasets. We analyzed five datasets to show how quantms can be utilized in the future of single cell proteomics. We thank the reviewer for their feedback which led us to expand quantms (<https://github.com/bigbio/quantms/pull/275>) to include data from timsTOF instruments.

3. It appears just the code is available without a public interface to use it. That means separate nodes would need to be set up for each user at their institution, requiring specialized expertise. I'm not sure how widely impactful this is if there is not a public interface.

R1.3 / In our documentation (<https://quantms.readthedocs.io/en/latest/>) we explained how to execute command line the workflow, in your local machine, or HPC cluster. nf-core and nextflow communities have made available a lot of documentation about how to run their workflows in AWS, Google Cloud and other on-premises cloud infrastructures. We have made available multiple YouTube training videos (quantms YouTube channel - https://www.youtube.com/playlist?list=PLMYykj5plyVeJZCJloWeo2f97K_gM_Cb) about how to use the workflow in English and Chinese:

- <https://www.youtube.com/watch?v=pBzelkqrPqQ>
- <https://www.youtube.com/watch?v=080me-EEVnU>
- https://www.youtube.com/watch?v=w3_bzzfSv7I

quantms is part of an ecosystem of workflows and pipelines (nf-core) that provide interfaces like tower (<https://cloud.tower.nf/>) that enable running the workflows using web interfaces. One example of how this will look is <https://nf-co.re/launch?pipeline=quantms&release=1.2.0> where parameters are translated into a web page interface. We are also part of the nf-tower default community showcase pipelines that you can easily run on free AWS trial credits when logging in for free into nf-tower (<https://tower.nf/orgs/community/workspaces/showcase/launchpad/260903829135591>)

While quantms can process small datasets in single computers (by using the conda or a container profile), the target audience is middle to large-scale research facilities that want to exploit the potential of HPC/cloud infrastructure for the fast, reproducible, and FAIR analysis of large scale-datasets. To address this concern of the reviewer, we will also include a detailed usage guide among the supplementary information for the paper.

Minor comments:

There are many typos and grammar errors throughout the text. For example, in supplementary note 1 "By February 2013", and the last two sentences in the abstract.

R1.4 / Thanks to the reviewer for this comment, we will correct all typos in the document.

Supplementary note 1 shows the different number of peptides from each search. Statistical tools exist to combine these outputs such as iprophet. Are the data not being combined in a statistically rigorous way? There should be another column for the combined peptide count.

R1.5 / We used the ConsensusID tool to combine PSMs from different search engines. Similar to iprophet, the previously published and proven ConsensusID (see https://abibuilder.cs.uni-tuebingen.de/archive/openms/Documentation/release/latest/html/TOPP_ConsensusID.html and <https://doi.org/10.1021/pr2002879>), combines search engine results probabilistically based on probabilities for each PSM. We support both SVM "probabilities" from Percolator (default) as well as PeptideProphet-like distribution-fitting-based probabilities. In that table, the peptides quantified correspond to the final number of peptides after the combination of both search engine results Comet and MSGF+. We will provide more explanations about the procedure itself and the report from the pmultiqc tool in the manuscript.

Link to the report mentioned in suppl note 1 doesn't work. The link to the PXD001819 benchmark doesn't work.

R1.6 / Thanks. This will be fixed in the document.

Supplemental notes line 128-129: "Protein sequence databases are made available in each dataset's GitHub folder." Is this statement correct? In my experience, it's rare that a dataset would store data in a github folder.

R1.7 / This sentence is not correct, as part of each reanalysis we are also including the fasta database used to perform the search. We will fix this in the manuscript.

Supplemental notes line 149-150: "the number of missing values in the background proteins was higher in quantms than MQ." This is not supported by the numbers in the table where missing values are lower than MQ (2.2% for quantms versus 31% for MQ; and 4.6% for quantms versus 5.3% for MQ).

R1.8 / We will correct this sentence in the document.

Supplemental note 1, figure 3: it says this data is from 48 proteins spiked into yeast, how can there be N>7,000 where yeast only expresses about 4k proteins? Also in the second panel showing the spike-in proteins, how can there be N>300 when there are 48 spike-in proteins?

*R1.9 / Thanks for pointing out that our figure legend is not sufficiently clear, and we will update the figure legend. The figure shows values summarized over all possible comparisons for a reference concentration. At a given reference concentration, all possible comparisons are included. For example, for reference concentration 50 fmol/ul, the comparisons 125~50, 250~50, 500~50, 2500~50, 5000~50, 12500~50, 25000~50, 50000~50 are considered. Consequently, there are at most 48*8 log2FC*

error values in the second panel and at most 1096*8 log₂FC error values in the first panel. Lower numbers indicate that proteins were not detected at certain concentrations.

Reviewer #2:
Remarks to the Author:

We think quantms is a potentially interesting tool with the novel concept of nextflow being implemented for proteomics analysis. However, there is a big question mark about whether such a tool could justify its publication at Nature Methods.

Here are some major critiques:

- All the software tools quantms integrated are not new. Also, nextflow was already published in Nature Biotech. Quantms is simply an assembly of published tools.

R2.1 / We recognize we did not do good work explaining why quantms is more than a collection of tools. We can split the development of quantms into these main categories of work:

- Experimental design-driven workflows allow to perform the data processing and downstream analysis in an automated way. Parallelization and distribution design of the tools: Most of the tools in the field of proteomics are monolithic, the same code-based is used to be run in the same machine. quantms design

(as mentioned by the reviewer) makes a clever design about how to parallelize these tools depending on the size of the experiment (number of raw files etc). For example, we parallelize the run of DIA-NN (DIA) for large-scale reanalysis of datasets (the main reason why Vadim Demichev is the author of the manuscript), this design enables us to reanalyze dataset PXD030304 within hours. The same approach is followed by the DDA workflow where multiple tools are carefully combined as small blocks that perform a specific task and can be replaced by new algorithms.

The quantms workflow is a software tool with over 14123 lines of code. It has logic to manage failed runs, memory-handling strategies, and file streaming. This logic is implemented using the Domain Specific Language (DSL) of nextflow. Nextflow is a programming language like Python, Java, or R.

- Novel development on existing tools: Our tools are now compatible with cloud and HPC environments. They have been ported to BioConda, allowing for easy deployment in these infrastructures. Popular tools like Comet, MSGF+, SAGE, DIA-NN, and OpenMS have been updated. Throughout this project, we have made significant improvements to multiple algorithms within the OpenMS tool ecosystem, including the proteomicsLFQ, ProteinInference algorithm, ProteinQuantifier, IDFilter, and MSstats exporter.
 - quantms offers tools to integrate parameters, reformat data, and perform ETL transformations. They export to standard file formats for easy reproduction and development. Multiple tools ensure consistency. QC is performed using the pmultiqc library, with interactive HTML reports sent after a successful pipeline run.
- It is impressive that the authors searched data from almost 10k human samples (>20k raw files) using quantms. This being said recent fast-searching tools such as MSFragger and pFind could easily search thousands of files in a single powerful PC. Also, the present large-scale searching was based on their own HPC cluster (204 cores, Supplementary Note 4), but not on a cloud resource like AWS. It is not clear if the cost range for such a search is too prohibitive for a normal user/lab using a cloud ecosystem.

R2.2 / We are grateful for this comment, first of all, while MSFragger and pFind are great tools, both of them are not designed for distributed computing for the following reasons:

- To use MSFragger, it must be manually downloaded from Michigan University and comply with their license. This makes it difficult to use in large-scale dynamic resource allocation architectures. MSFragger can't be put in open-source infrastructures like BioConda and Biocontainers limiting the reproducibility, scalability and automatic deployment of the tool with quantms.
- While individual projects like MSFragger/FragPipe have made great progress in optimizing their desktop tools to a level that they can be applied to larger datasets, FragPipe comes with a proven but rather fixed set of tools. It lacks flexibility when it comes to modifying or extending the built-in workflows. For example, integrating novel and alternative ML-based rescoring tools or replacing the search engine are important use cases for future quantms developments.
- COMET, the main search engine of quantms has similar performance in terms of CPU allocation and consumes less memory than MSFragger. pFind is a

Windows tool, that can't be used in most of the cloud and HPC infrastructures. As we mentioned before our main target audience/readers/users are researchers who want to perform the analysis in distributed architectures.

- Additionally, identification is only a small part of a large and robust protein quantification pipeline. The combination of identification, quantification, inference, QC and statistical post-processing needs to be carefully combined and tweaked to allow the robust processing of so many runs without errors.
- The core search engine of the workflow is COMET, which has been applied in some extensive studies, as evidenced by the recent Nature manuscripts [10.1038/s41592-022-01638-5, 10.1038/s41587-022-01539-0, 10.1038/s42255-022-00681-y, 10.1038/s41586-022-05314-8]. We do not think the quality of the results is affected by the search engine as demonstrated in our benchmarks, and research laboratories such as these previous examples can use quantms to carry out their analyses.
- To demonstrate the potential that quantms has to be extended and adapt to novel tools, we have integrated (during the review process) the new search engine SAGE which is 10 times faster than COMET and MS-GF+. The new release of quantms 1.2.0 included SAGE.

We understand that the reanalysis of so many datasets can't be performed by individual labs. We are using EBI infrastructure to perform such large reanalysis and provide the community with the final peptides/protein tables in a standard way that can be used by the community in different use cases:

- We integrate quantitative data from multiple datasets. Our ongoing studies have shown that claims made by independent studies can be easily disproven when integrated with a larger collection of datasets. Please note that this data has not yet been published.
- ML and Deep Learning communities can use the highly homogeneous data to generate and train new models.
- So far, we acknowledge, that we did a poor job of distributing the results of these large reanalyses in an easily accessible way. We have released a new web page quantms.org that centralizes all the datasets reanalyzed and we are looking to implement with PRIDE and MassIVE.quant to automatically push the data into those resources (as requested by reviewer 3).
- We provided an example to estimate the cost powered by nf-tower for the run of one experiment in AWS (<https://tower.nf/orgs/community/workspaces/showcase/watch/2UuiO5omjj8lm u>). The example is from an LFQ experiment, consisting of six raw files. The total running time was 18 minutes, with 0.4 CPU hours used and a total memory consumption of 6.87 GB. The cost estimation was 0.012 dollars. AWS is billed by time, but different clouds may have different billing methods.

We have found that errors such as file corruption and missing MS2 information can occur during data analysis. However, our quantms design allows for seamless continuation of the process from the point of failure. Our experience with MaxQuant and other tools has led us to believe that this feature, along with others, is crucial for the reanalysis of large-scale public quantitative data.

- No novel biological findings were present – it would be much more interesting if the author re-searched e.g., all the CPTAC data with quantitative outputs.

R2.3 / We fully agree that providing reanalysis data to the public is of greatest interest. At the current state, quantms, to our knowledge, already provide the largest dataset of proteins quantified in human healthy samples. While other resources like PeptideAtlas/GPMDB have been focused for years on protein identification, the current dataset provides the first large dataset of quantitative variation across samples, datasets, and tissues. We have already ongoing efforts to expand expression profiles over additional tissues and plasma where we found what proteins are consistently quantified in human plasma at which concentration.

However, we think the format of the manuscript, the journal and the aim are to probe that quantms is the only tool that currently out there can enable the systematic analysis of large collections of datasets in a reproducible way since the submission of the manuscript more than 110 additional datasets have been reanalysed and we plan to continue doing it and release the data to the public domain (<https://quantms.org>). It is also important that quantms is metadata-oriented, which means that the analysis starts from the SDRF (experimental design) and the output formats also contains the metadata information. For that reason, the main challenge is to be able to annotate the datasets and their corresponding metadata. We also agree that CPTAC is a highly interesting resource so important that we have annotated 66 CPTAC datasets (<https://github.com/multiomics/multiomics-configs/tree/master/projects/tumor>). We have started the process of reanalysing and we have submitted the first one to quantms.org PDC000126.

- The total number of 17k proteins identified, together with the caveat of not having a particular protein-level FDR control for the extremely large datasets, questioning such an impressive number of proteins for humans. Some benchmark work should be done by e.g., searching the data for non-human proteins or olfactory receptors (Ezkurdia et al 2014). Otherwise, such a title in NM could be misleading for the proteomics field.

R2.4 / Thanks to the reviewer for this comment. There is no question that proper protein-level FDR control is crucial for presenting extremely large datasets. Our approach is to use a unique workflow to analyze DDA (plex and LFQ)/DIA to provide quantitative information for proteins across numerous large-scale public datasets. The numbers reported were mainly intended to show the scale of our analyses. We have removed the total number of proteins quantified to avoid readers understanding the manuscript as a human quantified proteome and redirect the main focus of the integration and scalability of quantms reanalyses.

We are aware of the problems that large-scale integration prompts for the community and we have tried to address some of them in the following way (See also response 3.3):

- *For every project independent of the size, 1% FDR filters are applied at Peptide and Protein levels. Importantly, in some of the previous studies mentioned by the reviewer, Pandey*

- (<http://ftp.pride.ebi.ac.uk/pub/databases/pride/resources/peptomes/absolute-expression/PXD000561/>), the protein FDR was not applied. We have applied multiple files including peptide and protein FDRs and also some quality control measures from HPP guidelines for independent datasets.
- For the integrated/aggregated numbers of proteins per tissue/organ, we used some extra filters from HPP to guarantee that those proteins are the most reliable quantified across all experiments:
 - o Peptides should be higher than 7 AA in length.
 - o 2 unique peptides per protein (avoiding the indistinguishable grouping problem)
 - o Different to PeptideAtlas and other initiatives, we have collapsed the ms runs into the samples annotated by the users or from the manuscript. An enormous work has been performed to recover annotations about samples for each dataset, what PeptideAtlas consider an Experiment could be a replicate, or one fraction only. Instead, we grouped correctly the information at the sample level including replicates, technical or biological.

In addition to stringent FDR and QC control rules applied to independent reanalysis, we applied an adjusted protein FDR method for the integration build. While all of the protein inference procedures we utilized produce protein level q-values, not all of the report explicit decoy proteins with associated scores. To counter this, we introduced stand-in decoy protein entries for each protein list. The procedure injects stand-in decoys so that the ratio of decoy to target proteins corresponds to the target protein's q-values. After this operation, we merge the lists of proteins, now with stand-in decoy proteins, and sort the resulting list according to each protein's nominal q-values. In cases where there is more than one observation of a protein, only the protein with the lowest q-value was kept. We subsequently re-estimated each target protein q-value as the ratio of decoy to target proteins scoring as well or better than the protein.

Finally, we applied a strict protein-adjusted FDR threshold of less than 0.01 to filter the integration results. From the original quantified proteins with IBAQ (16336 tissues, 11403 cell lines and 5048 in plasma), the number of proteins that pass the 1% adjusted FDR are 16270 in tissues, 11374 cell lines and 4993 in plasma. A resource has been developed to quickly retrieve the IBAQ-based expression profile of the proteins quantified with quantms (Supplementary Notes 9, <https://quantms.org/ae>).

Finally, to validate our results to be biologically meaningful, we have double-checked olfactory protein IDs and none of them were found in the collection of the final quantified proteins.

- Finally, reading from the materials the author provided, we feel that the whole pipeline is not user-friendly, especially not for beginners. The troubleshooting of the

command lines can be painful and troublesome. No case study was present in detail. In general, the author could consider improving features (e.g., using GUIs) to facilitate the usage.

R2.5 / We agree with the reviewer that we can do a better job explaining how to run the workflow. Multiple videos have been added to the documentation and an explanation about how the workflow can be run using the nf-tower tool in AWS and other infrastructures. As a starting point for the reviewers, we suggest using the free 100 CPU hours on AWS via the easy-to-use web interface nf-tower: <https://tower.nf/orgs/community/workspaces/showcase/launchpad/260903829135591> (signup required). After evaluation, credentials for a cluster or cloud can be registered there. Besides that, it is as easy as:

```
# Install nextflow and nf-core
curl -s https://get.nextflow.io | bash
mv nextflow ~/bin/
pip install nf-core

# Launch the quantms pipeline
nf-core launch nf-core/quantms
```

to launch our pipeline directly on your computer or HPC cluster. You will be guided through a web-based interface with the parameters. The only required input is the SDRF and a path/URL to the protein database. We argue that this is even easier than most GUI-based programs out there.

To summarize, although we like the concept of this tool, we doubt how useful it could be for normal proteomic users and therefore the significance and novelty of this contribution.

R2.6 / We want to clarify why we believe that quantms is essential for the community. It's important to note that quantms isn't geared towards researchers or laboratories that employ tools like MaxQuant or ProteomeDiscover. Instead, it's designed to assist researchers, consortiums, and databases who need to carry out middle to large-scale studies and/or reanalyze public proteomics data. By utilizing quantms, we drastically reduced the processing time for dataset PXD010154 from one month to just four days, and for PXD030304, from several weeks to just a few hours. Additionally, quantms prioritizes resource allocation and can distribute raw files into smaller nodes with a low memory footprint, thereby avoiding the need to allocate an expensive machine for weeks.

On a separate note, it is not clear to us if quantms could support the direct, automatic procession of the PRIDE datasets (based on their IDs). If the normal users of PRIDE still need to download and copy the PRIDE datasets to a server, this can be very time-consuming (compared to a local analysis). If they do not need to, this is a highlight for quantms.

R2.7 / quantms supports direct, automatic processing of PRIDE datasets if the dataset was already annotated with an SDRF or the user provides his own SDRF. A URL to the SDRF can be passed as input to the workflow and the raw file URLs inside the

SDRF will be used to download each file into the respective step where they are needed. *quantms*' input is the newly adopted SDRF file format which contains the parameters and experimental design, *quantms* will perform the download automatically and perform the analysis. As soon as PRIDE supports stable links to these SDRFs accompanied with every dataset, it will be an extremely easy addition to our pipeline code to allow for a PRIDE accession number instead of a URL as the only input to our pipeline.

Minor:

1. The speed comparison to Maxquant was unfair due to the different HPC clusters used.

R2.8 / As the reviewer mentioned this comparison is unfair, the main reason why is not added in the main document of this discussion. MaxQuant and quantms follow different approaches, one (MaxQuant) needs a large node to perform all the analysis, and the other (quantms) can use multiple nodes of different sizes to perform the analysis. However, we think the plot and discussion can inform the readers of quantms how a dataset that was previously analysed in the best node available at EMBL-EBI in 1 month is now possible to be analysed in a few days using quantms. Depending on their architecture, and compute availability including cloud/HPC they can use quantms instead of MaxQuant.

2. It is not clear how *quantms* will incorporate and support different versions of software tools it currently assembles- This can be very important for, e.g., DIA-NN due to the development in the field of DIA data analysis.

R2.9 / quantms evolve with the tools that it integrates. When a new major release is done for a particular tool, a new release will be performed for quantms. Since the first release, multiple tools have been upgraded including OpenMS tools, Comet or DIA-NN. A robust testing system is in place that allows the developers of quantms to check for errors introduced in the tools included within quantms (see github actions - <https://github.com/bigbio/quantms/actions>). We also monitor other metrics for each tool such as execution time, and RAM. Our continuous integration system tests all workflows when new software, releases, or developments are introduced. If a user needs an older version and this version is available as a conda package or container, it is easy to overwrite the used version with a few lines in an additional config file. The final report of a run will include the exact versions used for each tool in this run for reproducibility (even if overwritten by the user).

3. Authors should cite MSfragger and pFind with some discussions on what factors the users should consider if they want to use *quantms*.

R2.10 / We will cite both tools in the main manuscript, and more discussion will be added about the target audience and which tool should be used in which specific case.

4. Figure 2B (run time and max memory usage) is not helpful because the final performance is linked to the particular HPC cluster they used and thus is not informative and transferable.

R2.11 / While we partly agree that runtime is hard to compare across CPU types with different clock speeds, max memory usage should be consistent across architectures with the same word size. Nonetheless, we think that both measures should help the reader to get a ballpark estimate of what to expect. We will clarify our intention with Figure 2B.

5. The TMT searching parameter cannot be found at https://nf-co.re/launch?id=1687271608_e665f052d31c

R2.12 / We have updated this URL and will provide a new one <https://nf-co.re/launch?pipeline=quantms&release=1.2.0> which presents how the workflow can be run from the nf-tower interface in AWS.

6. The link to Jupyter notebook <https://github.com/ypriverol/quantms-research/blob/main/notebooks/TMT/PXD007683Benchmark.ipynb> does not work.

R2.13 / We have fixed this URL in GitHub.

Reviewer #3:

Remarks to the Author:

The authors propose an open source, cloud-based pipeline for parallel reanalysis of quantitative proteomic datasets. The authors argue that the tool is needed due to an increasingly large sizes of the experimental datasets, which makes it hard to do the re-analyses on individual workstations.

While I agree that the tool is needed, I do not see a lot of novelty in the tool itself, as it is primarily an engineering effort that connects existing well-characterized modules. However, the datasets produced by a reanalysis with the tool are indeed novel and necessary. I believe that the manuscript will make an important contribution if it goes beyond stating the fact that we can re-analyze the studies on a cloud, and convinces the community that these are in fact reanalyses that we can trust. In my opinion the current version of the manuscript doesn't go quite far enough in that direction.

R3.1 / I appreciate the comments provided by the reviewers. In our response's introduction, we aimed to clarify why we consider quantms distinct from other tools. Rather than being a mere combination of existing tools, it represents a novel approach to parallelizing both algorithms and tools. We have developed new packages, tools, and libraries that enable reanalysis with an experimental design-aware approach.

As the reviewer pointed out, to our knowledge such a massive amount of quantitative proteomics data of healthy human samples with their corresponding sample annotations is very valuable to the proteomics community. We have created a web page <https://quantms.org> to centralize all the information about the quantms project including the documentation about the pipeline, the datasets reanalyzed and the absolute express protein builds (integrated view of the proteins across multiple tissues).

Currently, all the benchmark comparisons are in supplementary notes, and detailed explorations of reanalyzed datasets are limited to 3 controlled mixtures and 2 biological investigations. It is not clear how many plexes are in the TMT datasets, it appears that it is just one. The results may be quite different for multi-plex experiments. I could not locate a detailed example of a biological DIA dataset. The remaining datasets are mentioned in the passing. I suggest that the discussion of the reanalyzed datasets is moved front and center, and more evidence of accuracy of reanalyses is given for all the datasets, not just the selected few.

I would caution against focusing too much on controlled mixtures. While they are useful, they do not have biological variation representative of the actual studies. Therefore, I suggest giving more attention to data sets from human samples.

R3.2 / We agreed with the reviewer, that we will focus more on the discussion of the results of the reanalysis and their impact on the community. We will improve the explanation of the benchmark datasets in the supplementary information.

The number of identified proteins and the computational speed are of course important, but these are not the most important criteria in my opinion. Instead, I suggest that the authors focus on demonstrating the soundness of the quantitative analysis, and its accuracy. I have specific questions in this regard:

- How did the authors select dataset-appropriate settings at various stages of the analyses (such as choice of a database and modifications; mass resolution/accuracy; filtering out features with poor id or quant; filtering features with too much noise and too many missing values)? These criteria are dataset-specific. Is this decision automated? Or do users need to specify these parameters? What is the impact of the choice of these parameters on the quantitative accuracy? Which combinations result in more trustworthy reanalyzed datasets?

R3.3 / We are grateful for this feedback. As the reviewer pointed out, we have put in place a set of rules to perform each reanalysis, going from the size of the experiment (number of samples, replicates), etc (Please see response 2.4 also with details). We also gave priority to large-scale human sample datasets with more than 1000 raw files. Here is a table of the rules we have put in place to select datasets and some of the rules of our statistical framework (all of them are now included in the manuscript):

Dataset selection rules	
Rule	Comments
Dataset publication.	All datasets must be previously published in a scientific journal.
TMT, iTRAQ, DDA-LFQ, DIA-LFQ	Datasets were generated using one of the following analytical methods. quantms team has performed multiple studies to evaluate how intensity-based quantitation from plex, LFQ and DIA studies are comparable using quantms. References: [10.22541/au.168174437.77664121/v1, 10.1021/acs.jproteome.2c00812].

The following information was manually reviewed and annotated based on the original manuscript for each dataset:

- Instrument used.
- Artefactual Posttranslational modifications or enrichment performed for the sample.
- Tissue information.
- Cell type.
- Disease state: In this case we manually annotated for all experiments the healthy samples.
- Sample relation to files: All samples were correctly annotated including replicate information (biological/technical), fractions associated with each sample.
- Precursor and fragment tolerances: All precursor and fragment tolerances are annotated manually into an SDRF. The values were extracted from the original manuscript and refined using the following tool: param-medic [10.1021/acs.jproteome.7b00028]

All datasets are annotated in SDRF and can be found in the following repository in GitHub:

<https://github.com/multiomics/multiomics-configs/tree/master/projects/tissues>
 quantms uses these annotations automatically to set the settings for each file and each tool in each dataset appropriately

Peptide and protein identification rules

Rule	Comments
Protein database: Uniprot-Swissprot Reviewed database without isoforms.	We selected the Uniprot-Swissprot reference database to decrease the detection of features that can be reproduced across samples and decrease the impact in quantitation results of the shared peptides. We aim with this reanalysis's reproducible quant values across samples instead of more protein identifications.
Enzyme used: Trypsin	We reanalyzed only datasets which use trypsin as the enzyme (cleavage agent). As the previous rule, we aim to have features that are more reducible across samples and datasets. While the number of datasets from other enzymes are growing in the public domain, most datasets are based on Trypsin. This decision helps to get consistent iBAQ values for all datasets because the peptide spaces (tryptic space) are the same.
Dataset level PSM and Protein FDR: 1%	By applying a strict 1% FDR at PSM-level (before quantification) and protein(-group)

level on a final dataset-wide scale (after quantification) we guarantee the statistical control of the FDR for each dataset.

Peptide Length
We filter out all peptides less than 7 AA. This filter can be applied using the ibaipy library from quantms framework (<https://github.com/bigbio/ibaipy>). Future resources or views of the same data can apply more stringent filters, for example AA length > 9.

Number of unique peptides per proteins >=2
For a protein to be considered as reliably quantified, at least 2 unique peptides were needed.

Quantification rules

Rule	Comments
Number of samples per feature.	Every feature (Peptide + charge + retention time + replicate) must be present in at least 20% of the samples of each dataset.
Number of projects per protein and global FDR	We have a multi view approach for all the proteins quantified: - Dataset View: In this representation FDR is controlled at the level of the dataset as explained in previous section (1%). - Tissue proteomes View (https://quantms.org/ae): To achieve this, we began by merging all protein lists with protein q-values from various datasets, while keeping them separated based on the dataset type (e.g. cell lines and tissue). Next, the algorithm generated a distribution of decoy proteins that were similar to the target proteins in the integration list. The protein-adjusted FDR was calculated based on this distribution of decoy proteins. Finally, we applied a strict protein-adjusted FDR threshold of less than 0.01 to filter the integration results.

Data provenance and QC

Rule	Comments
All results must be in standard file formats as follows: - Sample metadata: SDRF - Spectra: mzML Peptide/Protein identifications: mzTab Peptide, Protein: scores	All the results from these reanalyses are in standard file formats. All peptide and protein scores including posterior error probabilities, p-values and search engine scores are available.

pmultiqc
<https://github.com/bigbio/pmultiqc>

*This can help the community and future resources to detect manually low-quality signals (peptides and protein identifications/quantitation values). Quality control reports for every dataset are provided using the newly developed library *pmultiqc* (<https://github.com/bigbio/pmultiqc>). The library well explained in the manuscript and the *quantms* documentation provides multiple plots and statistics to detect and visualize problems in the quantitative results.*

- How can the tool check that the order of the acquisitions is appropriately randomized? Can the users be warned that the results cannot be trusted due to lack of or inappropriate randomization?

*R3.4 / We don't have control over how data was acquired so we need to rely on our quality control plot to reveal errors in experimental design annotations or e.g. drop in instrument performance. More subtle errors, e.g., caused by improper randomizations – as pointed out by the reviewer, can easily lead to suboptimal results in individual studies. Whether these can be automatically recognized, and users warned is to our best knowledge an active area of research. We would love to learn about reliable methods so that these can be integrated into *pmultiqc* in the next release.*

- How do the authors choose the between-run normalization and correction for batch effects? For example, median normalization may not be appropriate for some controlled mixtures, or for experiments with standards. If the experiment includes standards, can/should the normalization be done with respect to these standards? These decisions are dataset-specific. Are they automated? What is the impact of the choice on the accuracy?

R3.5 / Thanks to the reviewer for this question. This is how the quantitative values are computed:

1. First, for the intensity-based absolute expression data integration, no mixtures are used. Mixture datasets were only used in the manuscript to demonstrate the accuracy of the pipeline in obtaining intensity values similar to MaxQuant. For normalization of iBAQ values, we use reliable features (as explained before) and normalization is done by sample using the rIBAQ previously used by multiple studies [10.1038/s41597-021-00890-2, 10.1038/sdata.2018.128]. Currently, no normalization is performed against standards since they are not available in the vast majority of studies.

During the first release of this data (this study), we aimed to produce reproducible iBAQ values for all the reanalyses. However, we have already observed during integration for high-quality integration, not all the datasets could be integrated, and

automatic integration is not possible. For example, here (next figure) is a correlation plot of all proteins quantified in plasma (data included in this paper, but not the biological and downstream curation):

It can be easily observed that PXD009219 has a low correlation compared to any of the reanalyses. Then, integration like these helps to validate and verify/revoke claims on independent datasets.

In a second iteration and possible resource, we aim to produce core proteomes (well-curated) integrated datasets where not only quantitative values are presented but also expert manual inspection can help to curate the data. We aim to use the current data in this study and analyze use-case by use-case: Plasma proteome, heart proteome, etc. As we mentioned before the extended proteome data (released here) will be extended into core proteome views including better normalization/imputation methods.

- How are the reanalyses made available to the community, at the intermediate and at the final stages? E.g., if one wanted to change only one module in the workflow for one of these datasets, can we start from an intermediate step, or do we need to restart the analysis from scratch? If we can start in the middle, where can we find the intermediate input files?

R3.6 / Thanks to the reviewer for this question. First of all, intermediate files are kept for some of the steps for provenance and reproducibility and can be downloaded by external users. We currently do not share the entire folder of the runs but only the output of most important steps as well as the final results, but this is a matter of internal configuration and could be changed if this becomes a prominent use case. Running and resuming workflows can be performed by users with appropriate access rights to the internal file system and compute infrastructure (e.g., in our case EBI personnel). At EBI, we currently, do not provide a public interface for changing and re-running

analyses on PRIDE's infrastructure as it comes with some security implications. However, all quantms modules can be replaced by anyone if run locally and even resumed if the entire run folder is downloaded. A resumed run will start with the steps corresponding to the replaced modules (and everything downstream). One of its strengths of using nextflow in quantms is that the workflow is highly flexible and configurable, and all results can be exported and shared with the community, however, we tried to make a sensible choice of what to export and want to see how the data is used, and what data is really demanded by the community in order to decide which modules' intermediate results need to be shared in addition. For example, currently, quantms is starting to be used in more research studies and we are also interested to share with the community not only the quantified peptides/proteins but potentially also features without peptide identifications. For that reason, we are also exporting all the features in the consensusXML files (e.g., http://ftp.pride.ebi.ac.uk/pub/databases/pride/resources/proteomes/absolute-expression/PXD000561/proteomicslfg/PXD000561.sdrf_openms_design_openms_consensusXML). We hope that it is understandable that the amount of intermediate data to be stored and made available heavily impacts maintenance and hardware costs at PRIDE and therefore needs to be considered carefully.

- Since the workflow is modular, and some modules can be substituted, could the authors illustrate the impact of this modularity on the quantitative results? I.e., how similar or different are the results when you substitute one module with another module designed for the same purpose?

R3.7 / We really appreciate the reviewers for these questions. First, one of the strengths of quantms is indeed its modularity. For example, during the review process, we integrated as a module the new search engine SAGE (discussion added in the Supplementary Notes). The quantms community will always benchmark the impact of the new modules on the results. Prior to its use as the default tool for certain tasks, specific components of quantms have been benchmarked, such as the differential expression step (referenced in [10.1021/acs.jproteome.2c00812]).

*Modules can be selected using parameters, for example, search engines can be selected independently or combined using the parameter **search_engines**. In the future, other modules such as re-scoring tools, or other differential expression analysis tools can be integrated in the workflow.*

quantms is an open-source project, we coordinate the integration of new modules over multiple channels: github, slack channel. All integrations are followed by benchmarks and follow-up public discussions. For the data released in the present manuscript: if new modules are added or changed, we will certainly benchmark and evaluate the impact on ground-truth datasets first and if deemed significant enough, a re-analysis of affected studies will be performed. We have already developed downstream scripts and packages including `pmultiqc` to evaluate the impact of each change. To hypothesize about the impact of a specific change is in our opinion very difficult. It heavily depends on which step is changed and how different the used algorithm is from the one being replaced.

- For previously published data sets, how similar or different are the results as compared

to the original publications, e.g., in terms of number of proteins, features per protein, missing values, (relative) protein quantification or detection of differentially abundant proteins? What can explain the differences, if any? Which analysis is more trustworthy?

R3.8 / Table 6 includes now a new column with the number of peptides and proteins quantified originally for each dataset and by quantms. From our recent research [10.1021/acs.jproteome.2c00812], we don't observe major differences between MaxQuant and quantms. However, we observed major differences between quantms and other common tools like ProteomeDiscover, or Mascot compared with quantms. However, our aim is to reanalyze as many as possible datasets with the same workflow and make this data available. The value of these reanalyses is the following:

- *The same tools and versions, scores, and data formats are used to generate the data in controlled execution environments. This yields highly reproducible and consistent results which are important for downstream analysis and comparison between many experiments. Additionally, all the sample metadata associated with the results will be provided for the users.*
- *In some specific cases, more proteins are quantified with quantms compared with other workflows.*
- *QC reports and results in standard file formats are provided for all datasets facilitating the flourishing of use cases like expression pattern recognition, ML model development, and ultimately understanding how protein expression works across multiple datasets.*

Additional questions:

What is the relationship between the annotation formats used by this framework, annotation formats in MassIVE.quant, and annotation formats for quantitative experiments in PRIDE? What is the relationship between quantms and the reanalysis capabilities in MassIVE? Can the intermediate analysis files be stored in MassIVE.quant?

R3.9 / To perform the reanalysis, we used the SDRF annotation format [10.1038/s41467-021-26111-3], which was created in partnership between ProteomeXchange and HUPO-PSI (<https://psidev.info/srdf>) and the major partners PRIDE, MassIVE and PeptideAtlas. SDRF can be used for submissions in PRIDE and MassIVE.quant, allowing for automatic reanalyses using quantms without needing the curation team to re-annotate the datasets. As of December 2023, over 200 submissions have been made in PRIDE using SDRF.

MassIVE.quant supports reanalyses performed by different tools and users. Similar to previous approaches like PeptideShaker [10.1038/nbt.3109] (with PRIDE collaboration), the idea is to enable users to reanalyse datasets and deposit them back to archives like MassIVE.quant or PRIDE. While both approaches promote the reanalysis of public data, in our opinion their goal has been limited by the following challenges:

1- The process of reanalyzing quantitative data is hampered by current computing and approaches, particularly with MaxQuant. In 2020, MassIVE.quant was launched with

95 datasets, which has only increased to 114 datasets by 2023 (equivalent to three datasets per month). *quantms* data, on the other hand, was reanalyzed in just four months, involving 91 datasets. Since the publication of the original study (2 month ago), over 40 datasets have been reanalyzed, and 60 CPTAC datasets are now being analyzed for release. The datasets released in this study (with an average of 371 files per experiment) contain larger DIA/DDA/DDAplex datasets compared to *MassIVE.quant*, which only has an average of 11 files per reanalysis. Most of the reanalyses in *MassIVE.quant* involve method/benchmark datasets, while the datasets in this study are all biologically relevant.

The quantitative reanalysis of the following four large studies [PXD000561, PXD000865, PXD010154, PXD016999] hasn't previously been performed. We think that *quantms* is the key point here by enabling parallelization, memory low footprint, and re-launching and automatic reallocation of the jobs.

2- Lack of metadata. All datasets released within this study are annotated with the corresponding sample metadata using the standard file format SDRF [10.1038/s41467-021-26111-3]. We have annotated all the datasets provided *quantms* including in all the exported datasets the corresponding metadata for the downstream analysis (e.g., *MSstats* input).

We have contacted *MassIVE.quant* team and transferred the first reanalysis to it. However, we are building with *MassIVE.quant* a protocol and pipeline to automatically transfer reanalyses without the need to use the user interface. This is possible because *MassIVE.quant* reanalyses are based on *MSstats* input, *mzTab* and *mzML* files, the same output of *quantms* pipeline. In the meantime, we have created a webpage <https://quantms.org/datasets> listing all the reanalysis performed with *quantms*.

Decision Letter, first revision:

Dear Yasset,

Thank you for submitting your revised manuscript "quantms: A cloud-based pipeline for quantitative proteomics enables the reanalysis of 13132 human samples." (NMETH-BC52562C). It has now been seen by the original referees and their comments are below. The reviewers find that the paper has improved in revision, and therefore we'll be happy in principle to publish it in Nature Methods, pending minor revisions to satisfy the referees' final requests and to comply with our editorial and formatting guidelines.

TRANSPARENT PEER REVIEW

Please note: we allow redactions to authors' rebuttal and reviewer comments in the interest of confidentiality. If you are concerned about the release of confidential data, please let us know specifically what information you would like to have removed. Please note that we cannot incorporate redactions for any other reasons. Reviewer names will be published in the peer review files if the reviewer signed the comments to authors, or if reviewers explicitly agree to release their name. For more information, please refer to our FAQ page.

ORCID

Sincerely yours,
Allison

Allison Doerr, Ph.D.
Chief Editor

Nature Methods

Reviewer #1 (Remarks to the Author):

I appreciate that the authors spent effort toward my main comments, which related to showing some biological benefit, application to single cell proteomics data, and the availability of the tool. I'm confident that they did a lot of work and their approach is valid and useful. Their conclusions are likely valid as well. Unfortunately, after reading all the reference reports, it's still not clear to me how this platform will be widely adopted or that quantifying these additional proteins would impact biological conclusions. These issues combined with a lack of novelty in the algorithms make me feel that this manuscript would easily find a more appropriate home in a journal with more specific readership.

Reviewer #2 (Remarks to the Author):

The authors have made substantial and high-quality revisions to improve their tool, quantms. They have also responded to and addressed most of our questions and suggestions. Although some issues with novelty remain, we believe that quantms represents a significant advancement in enabling the systematic analysis of large datasets in a reproducible manner. It would be beneficial to the community if the authors continue to release new data and updates to the public domain.

The authors have put considerable effort into enhancing the FAIR principles with the creation of <https://quantms.org>. I would suggest a following minor revision so that they can improve the Usability and user experience.

For instance, the interface of NF-Tower of <https://tower.nf/orgs/community/workspaces/showcase/launchpad/260903829135591> seems lots of parameters have to be set up by the users, and it is not so easy-to-use as claimed. Unfortunately, using the default parameters results in a "failed" warning. Moreover, the process for local installation also demands the configuration of lots of parameters that are not easy to use.

Here are some additional minor suggestions:

- The "Absolute" page on QuantMS.org could be a valuable resource for proteomics researchers and other biologists. Some minor improvements could enhance its utility: a) Provide a brief explanation of iBAQlog (log2 or log10?). b) Enable the comparison of multiple proteins (e.g., up to five proteins) using boxes of different colors. c) Include the number of observations (n) for each box, alongside Q1 and Q3. d) Allow for the use of gene symbols in addition to UniProt identifiers.
- The font sizes in the main figures are too small in many places. Optimizing the visualization would improve readability.
- Please cite the papers associated with the PXD numbers in the main text.

Reviewer #3 (Remarks to the Author):

I would like to thank the authors for the revisions. I am happy to see that the authors took my comments seriously, and included details of reanalyses of more datasets. In my opinion the revisions substantially improved the impact of the work.

However, I was unable to fully appreciate the results of the reanalyses, for two reasons.

First, the font of the figures in both main and supplementary is too small, and even on my very large screen I was unable to read the content of the figures. The authors should revisit the figures so that we can understand the details.

Second, I am not quite sure what to conclude from the reanalyses, besides the fact that they could be done. The authors did a good job summarizing various metrics. However I am not sure what these metrics imply. Are the reanalyses consistent enough with the analyses in the original manuscripts? Is the quality of identification, and in particular of the quantification, acceptable for downstream biological investigation? How can we use these reanalyses to gain biological insights beyond what was done in the original studies? I understand that discovering new biology from these reanalyses is outside the scope of the manuscript. However, I would really appreciate some discussion of how these reanalyses can be used by broader scientific community, and of the evidence that the reanalyses are of a good enough quality to support meaningful downstream use.

Author Rebuttal, first revision:

Thanks to the reviewers for their comments and feedback. We have addressed all of the reviewers' comments below to improve the quantms manuscript and the website.

Reviewer #1:

Remarks to the Author:

I appreciate that the authors spent effort toward my main comments, which related to showing some biological benefit, application to single cell proteomics data, and the availability of the tool. I'm confident that they did a lot of work and their approach is valid and useful. Their conclusions are likely valid as well. Unfortunately, after reading all the reference reports, it's still not clear to me how this platform will be widely adopted or that quantifying these additional proteins would impact biological conclusions. These issues combined with a lack of novelty in the algorithms make me feel that this manuscript would easily find a more appropriate home in a journal with more specific readership.

R/ We appreciate that the reviewer finds useful and valid quantms results and approaches.

- *We want to stress that the algorithms and parallelization used in quantms are innovative and groundbreaking in the field of computational proteomics. For years, desktop*

applications have dominated the field, analyzing data on a single node. Although attempts have been made in the past to analyze cluster infrastructures, quantms is the first solution that enables quantitative analysis using multiple analytical approaches. With the parallelization of DIA workflow and corresponding algorithms, we have successfully reanalyzed a dataset (PXD039236) that includes over 15,000 raw files in library-free mode, which was not possible before using any DIA tool. With sufficient computational resources, quantms have the potential to become one of the leading tools for clinical and large-scale analysis of proteomics datasets.

- quantms is a unique solution that utilizes experimental design and sample metadata in a standard file format as the input and leading format for the entire analysis. This may seem like a minor detail, but it is a significant improvement that will enable the proteomics community and other tools to move towards more metadata and reproducibility of the results. Additionally, quantms exports all results in standard file formats, which is a first in the field. This feature will enable better reuse of the results obtained.
- Related to the data, here are some of the potential uses that we have started seeing in quantms data:
 - Most of the reanalysis datasets and resources that perform reanalysis of data are identification-oriented (PeptideAtlas, MassIVEKB, etc). However, the data generated with quantms is the first collection of curated datasets from human healthy tissues, and cell lines where users can find what specific protein is expressed on which particular tissue and what is the number of samples in which this expression has been seen. With quantms, you can do those queries and also compare different expression patterns for multiple proteins which could be helpful for biologists studying multiple proteins (targets).
The quantms team is now in conversations with the UniProt team to use the quantms data to improve annotations of the expression specificity field in UniProt. UniProt has a field called tissue specificity where it defines where a particular protein has been seen expressed and what is the frequency of this expression (how many samples), for a long time UniProt has been using manuscript annotations but now quantms could help to refine the manual annotation process.
 - Quantms data expression profiles could be downloaded in CSV format for all the projects including all the sample annotations. Multiple research groups and companies are already working with the expression profiles and the annotations to develop new machine-learning algorithms to predict protein co-expression networks; refine batch effect correction methods using deep-learning algorithms and predict protein specificity expression using also AI methods. As more data becomes available in quantms with the corresponding annotations and sample metadata, more use cases will be growing similar to other fields such as transcriptomics or genomics.
 - The quantms team has started already to annotate and reanalyse single-cell datasets which will trigger the development of new algorithms to combine “bulk” tissue expression profiles with single-cell data. In addition, more data is being reanalyzed and released to the public for phosphoproteomics and other posttranslational modifications.

Reviewer #2:

Remarks to the Author:

The authors have made substantial and high-quality revisions to improve their tool, quantms. They have also responded to and addressed most of our questions and suggestions. Although some issues with novelty remain, we believe that quantms represents a significant advancement in enabling the systematic analysis of large datasets in a reproducible manner. It would be beneficial to the community if the authors continue to release new data and updates to the public domain.

The authors have put considerable effort into enhancing the FAIR principles with the creation of <https://quantms.org>. I would suggest a following minor revision so that they can improve the Usability and user experience.

For instance, the interface of NF-Tower of <https://tower.nf/orgs/community/workspaces/showcase/launchpad/260903829135591> seems lots of parameters have to be set up by the users, and it is not so easy-to-use as claimed. Unfortunately, using the default parameters results in a "failed" warning. Moreover, the process for local installation also demands the configuration of lots of parameters that are not easy to use.

R/ We agreed with the reviewers that the adoption of cloud workflows is more complex than what most users are familiar with. The nf-core team has a lot of documentation on how to run a workflow in nf-tower (<https://docs.segера.io/>), videos (<https://www.youtube.com/@Nextflow/videos>) and a lot of support (<https://community.segера.io/>). The quantms channel in nf-core is used for training new users about how to start with the pipeline (<https://nfcore.slack.com/channels/quantms>).

The local installation as far as we know only demands the installation of docker/singularity or bioconda which will resolve the dependencies of the tools, and the nextflow as the workflow engine. The required parameters (minimum required) is a FASTA protein sequence database, the RAW files and the SDRF which contains the experimental design and the sample metadata. Most of the parameters have default values which are commonly used by most of the proteomics experiments (1% FDR and Peptide and Protein level), Enzyme Trypsin, etc. The quantms team

continue working to develop algorithms that enable users to run the workflow with fewer parameters, including the removal of fragment and precursor tolerances which currently are needed for both DDA approaches DDAplex and LFQ methods.

We have made available in quantms.org all the SDRFs for each of the analyses and the command line used to perform the analysis. For example:

- SDRF https://ftp.pride.ebi.ac.uk/pub/databases/pride/resources/peptomes/absolute-expression/MSV000079033.1/pipeline_info/MSV000079033-Blood-Plasma-TMT6.sdrf.tsv
- Commandline https://ftp.pride.ebi.ac.uk/pub/databases/pride/resources/peptomes/absolute-expression/MSV000079033.1/pipeline_info/execution_report_2023-05-11_10-33-04.html

*We are committed to providing support and assistance to [quantms](https://quantms.org) users, especially our target audience which includes core bioinformatics facilities, bioinformatics cloud infrastructures, and laboratories analyzing large-scale experiments. Furthermore, we are actively working on reducing the number of parameters and steps required to install the workflow for desktop users who wish to run the workflow on their machines. It's worth noting that [quantms](https://quantms.org) is a workflow that forms part of the *nf-core* community, which releases guidelines for running all analyses following certain protocols that improve reproducibility and data analysis transparency. This implies that [quantms](https://quantms.org) and *nf-core* enforce certain best practices, such as resolving dependencies over containers and packages, which must all be installed independently with their corresponding versions. This ensures that the same analysis can be re-run in the future with the given data.*

Here are some additional minor suggestions:

- The "Absolute" page on [QuantMS.org](https://quantms.org) could be a valuable resource for proteomics researchers and other biologists. Some minor improvements could enhance its utility: a) Provide a brief explanation of iBAQlog (log2 or log10?). b) Enable the comparison of multiple proteins (e.g., up to five proteins) using boxes of different colors. c) Include the number of observations (n) for each box, alongside Q1 and Q3. d) Allow for the use of gene symbols in addition to UniProt identifiers.

R/ We thank the reviewer for these comments we have implemented all of them. Here are some examples:

- Searching by gene name is now possible, multiple proteins could be compared (up to 5): <https://quantms.org/ae/tissues?protein=LRBA&protein=PGAM5&protein=LRP8>

- We have added a link to the *ibaqpy* package which is the algorithm and tool that we use to compute the *ibaqpy* values (<https://github.com/bigbio/ibaqpy>). Additionally in the repo of *ibaqpy* a manuscript where *ibaq* values are benchmarked between TMT and LFQ experiments.

- The font sizes in the main figures are too small in many places. Optimizing the visualization would improve readability.

R/ The figures have been refined in the present version.

- Please cite the papers associated with the PXD numbers in the main text.

R/ We will collaborate with the journal to provide accession numbers for all datasets. The supplementary information includes a table with the complete list of datasets.

Reviewer #3:

Remarks to the Author:

I would like to thank the authors for the revisions. I am happy to see that the authors took my comments seriously, and included details of reanalyses of more datasets. In my opinion the revisions substantially improved the impact of the work.

However, I was unable to fully appreciate the results of the reanalyses, for two reasons.

First, the font of the figures in both main and supplementary is too small, and even on my very large screen I was unable to read the content of the figures. The authors should revisit the figures so that we can understand the details.

R/ We will work with the journal to improve the figure's quality. The current version of the manuscript and supplementary information have better quality on each figure.

Second, I am not quite sure what to conclude from the reanalyses, besides the fact that they could be done. The authors did a good job summarizing various metrics. However I am not sure what these metrics imply. Are the reanalyses consistent enough with the analyses in the original manuscripts? Is the quality of identification, and in particular of the quantification, acceptable for

downstream biological investigation? How can we use these reanalyses to gain biological insights beyond what was done in the original studies?

R/ We would like to clarify and discuss how we currently use the generated data. The UniProt team is collaborating with the quantms team to refine the annotation of each protein's tissue specificity. This annotation helps users understand where a given protein is more frequently expressed, which can be useful for biologists and researchers when deciding if the protein is a suitable target for a particular drug. This particular annotation is now manually annotated from manuscripts and the UniProt team would like to complement it using the quantms expression profiles.

quantms has recently become the second database to offer expression profiles for human data in multiple tissues, similar to proteomicsDB. However, while proteomicsDB only provides data from DDA-LFQ experiments, quantms has added a lot of data from DIA datasets which complements the existing data from proteomicsDB. Different from the transcriptomics field where multiple leading resources such as ExpressionAtlas (<https://www.ebi.ac.uk/gxa/home>), GEO profiles (<https://www.ncbi.nlm.nih.gov/geoprofiles/>) or bgee (e.g. <https://www.bgee.org/gene/ENSG00000174948#expression>) offer well-annotated gene expression profiles; in proteomics researchers lack of multiple resources that contains expression profiles of proteins in multiple tissues, conditions and diseases. Only proteomicsDB and HPA (which is not mainly MS-based) provide this information, we expect that quantms data can help users to understand more protein expression across tissues, cell lines and diseases (data still not public in the resource but already processed with quantms). Reanalysis with the same pipeline (quantms) not only allows the reanalysis of large-scale datasets (as presented in the manuscript) but also allows to integration of results from multiple datasets. Similar to the CPTAC consortium, where data gets annotated and analyzed with similar workflows, we aim to analyze data in the public domain with the same workflow, UniProt reference proteome and more importantly to annotate the datasets with enough information that enables users to reuse these protein expression profiles in their analysis.

We have made the data available in multiple ways:

- *With a web interface for users that just one to check the expression of a given protein or set of proteins (e.g. <https://quantms.org/ae/tissues?protein=LRBA&protein=PGAM5&protein=LRP8>).*
- *Independent datasets are provided for the users. It is important to notice that a lot of these datasets have been previously submitted to ProteomeXchange and the repositories as PARTIAL submissions, which means that the expression data is not available only the RAW data (e.g. MSV000079033 - <https://massive.ucsd.edu/ProteoSAFe/dataset.jsp?task=9c3f5d8472c1486a8fceda556598ac94>), in addition, quantms team has reannotated them extensively to enable users to have access to the sample metadata.*

Is the quality of identification, and in particular of the quantification, acceptable for downstream biological investigation? How can we use these reanalyses to gain biological insights beyond what was done in the original studies?

R/ We agreed with the reviewer that the identification and particular quantification are not the only factors relevant for downstream biological investigations and future use of the quantms data. Our aim with all the benchmarks we made in the study is to make clear that the results of the workflow are correct and comparable with other tools such as MaxQuant and ProteomeDiscover.

However, we believe that the main difference lies in the quality and quantity of the data we are providing, as well as the annotations provided by quantms analyses.

- *All the samples for every dataset reanalyzed with quantms have been annotated using ontologies (EFO), which means that for example every liver sample or heart left ventricle. This level of annotation is not available in any proteomics resource which limits the possibility of integrating or reducing the capabilities to discover new biological insights. For example, the quantms team is now building a heart proteome within quantms to make it available in the resource which will enable users to detect different expression profiles, not only at the level of the tissues but also regions of the organs.*
- *quantms is scalability enable us to release to the public builds with millions of peptides, and thousands of proteins quantified and reanalyzed the major datasets in the public domain at large scale. Users will have access to valuable protein expression profiles for multiple tissues, diseases, and species by accessing these datasets. For example, because of the novelty and parallelization capabilities of quantms, we have recently managed to reanalyse the largest plasma quantitative proteome experiment (PXD039236) with more than 15'000 ms runs. By integrating this dataset with more than 40 datasets already available for plasma, we have quantified more than 7000 proteins in plasma, more than 3000 never seen (quantified) before (data not shown).*

To summarize, our ambition is to facilitate new biological discoveries and claims not only based on the quality of protein identifications but also on the amount of data integrated from different diseases, tissues, and cell lines. Furthermore, the manually annotated datasets using ontology terms will provide a unique resource for the development of new AI algorithms and other resources that will consume data from quantms.

I understand that discovering new biology from these reanalyses is outside the scope of the manuscript. However, I would really appreciate some discussion of how these reanalyses can be used by broader scientific community, and of the evidence that the reanalyses are of a good enough quality to support meaningful downstream use.

R/ We agreed with the reviewer that some details about the future use cases and how much can be done with the data should be included in the manuscript. We have added in the conclusions of the manuscript a discussion about possible use cases and integration with other resources.

Final Decision Letter:

Dear Yasset,

I am pleased to inform you that your Brief Communication, "quantms: A cloud-based pipeline for quantitative proteomics enables the reanalysis of public proteomics data.", has now been accepted for publication in Nature Methods. The received and accepted dates will be 12 May 2023 and 3 June 2023. This note is intended to let you know what to expect from us over the next month or so, and to let you know where to address any further questions.

Over the next few weeks, your paper will be copyedited to ensure that it conforms to Nature Methods style. Once your paper is typeset, you will receive an email with a link to choose the appropriate publishing options for your paper and our Author Services team will be in touch regarding any additional information that may be required.

Once proofs are generated, they will be sent to you electronically and you will be asked to send a corrected version within 48 hours. It is extremely important that you let us know now whether you will be difficult to contact over the next month. If this is the case, we ask that you send us the contact information (email, phone and fax) of someone who will be able to check the proofs and deal with any last-minute problems.

If, when you receive your proof, you cannot meet the deadline, please inform us at rjsproduction@springernature.com immediately.

Please note that *Nature Methods* is a Transformative Journal (TJ). Authors may publish their research with us through the traditional subscription access route or make their paper immediately open access through payment of an article-processing charge (APC). Authors will not be required to make a final decision about access to their article until it has been accepted. Find out more about Transformative Journals

Authors may need to take specific actions to achieve compliance with funder and

institutional open access mandates. If your research is supported by a funder that requires immediate open access (e.g. according to Plan S principles) then you should select the gold OA route, and we will direct you to the compliant route where possible. For authors selecting the subscription publication route, the journal's standard licensing terms will need to be accepted, including self-archiving policies. Those licensing terms will supersede any other terms that the author or any third party may assert apply to any version of the manuscript.

If you are active on Twitter/X, please e-mail me your and your coauthors' handles so that we may tag you when the paper is published.

To assist our authors in disseminating their research to the broader community, our SharedIt initiative provides you with a unique shareable link that will allow anyone (with or without a subscription) to read the published article. Recipients of the link with a subscription will also be able to download and print the PDF. As soon as your article is published, you will receive an automated email with your shareable link.

Please note that you and your coauthors may order reprints and single copies of the issue containing your article through Springer Nature Limited's reprint website, which is located at <http://www.nature.com/reprints/author-reprints.html>. If there are any questions about reprints please send an email to author-reprints@nature.com and someone will assist you.

Best regards,
Allison

Allison Doerr, Ph.D.
Chief Editor
Nature Methods